# Realization of active metamaterials with odd micropolar elasticity

Yangyang Chen [1,5], Xiaopeng Li[1,5], Colin Scheibner [2,3,5], Vincenzo Vitelli [2,3,4✉] & Guoliang Huang [1✉]

Materials made from active, living, or robotic components can display emergent properties arising from local sensing and computation. Here, we realize a freestanding active metabeam with piezoelectric elements and electronic feed-forward control that gives rise to an odd micropolar elasticity absent in energy-conserving media. The non-reciprocal odd modulus enables bending and shearing cycles that convert electrical energy into mechanical work, and vice versa. The sign of this elastic modulus is linked to a non-Hermitian topological index that determines the localization of vibrational modes to sample boundaries. At finite frequency, we can also tune the phase angle of the active modulus to produce a direction-dependent bending modulus and control non-Hermitian vibrational properties. Our continuum approach, built on symmetries and conservation laws, could be exploited to design others systems such as synthetic biofilaments and membranes with feed-forward control loops.

[1] Department of Mechanical and Aerospace Engineering, University of Missouri, Columbia, MO 65211, USA. [2] James Franck Institute, The University of Chicago, Chicago, IL 60637, USA. [3] Department of Physics, The University of Chicago, Chicago, IL 60637, USA. [4] Kadanoff Center for Theoretical Physics, The University of Chicago, Chicago, IL 60637, USA. [5]These authors contributed equally: Yangyang Chen, Xiaopeng Li, Colin Scheibner. ✉email: vitelli@uchicago.edu; huangg@missouri.edu

Responsive materials in both biology and engineering distinguish themselves by their ability to respond to external stimuli in tailored ways[1–7]. For example, muscles contract in response to electrical signals[8] and mechanocaloric solids undergo dramatic deformations in response to temperature changes[9]. Unlike computers or multicellular organisms with specialized functional components, the undifferentiated physical machinery available in a distributed material presents unique challenges for sensing, information processing, and response. Yet, the range of available functionalities is fundamentally extended when the materials possess distributed, local reservoirs of energy[4,10–19]. Such active materials exhibit responses not allowed by their passive, or energy conserving, counterparts.

Activity is intimately connected, but not equivalent, to a family of material symmetries collectively known as reciprocity[20–28]. Here we distinguish between two notions of reciprocity relevant for the design of mechanical metamaterials. The first notion is a generalization of Newton's third law, which states that the forces between any two components of a mechanically isolated system must be equal and opposite. This notion of reciprocity can be formulated for other generalized momenta, such as angular momentum. Mechanical systems that violate this version of reciprocity must fundamentally be in contact with an external medium, such as a substrate or a background fluid, to act as a momentum sink. For systems described by a Lagrangian or Hamiltonian, translational (or rotational) symmetry gives rise to conservation of linear (or angular) momentum. Hence, systems with translational symmetry that violate this first notion of reciprocity must either be dissipative, driven, or active.

A second conceptually distinct notion of reciprocity is known as Maxwell-Betti reciprocity, which can be roughly defined as the symmetry between perturbation and response. When a mechanical system is deformed, the work done can schematically be written as $dW = \sum_a \sigma_a du_a$. Here, $u_a$ is a short hand notation for mechanical degrees of freedom such as displacements, rotations or strains, and $\sigma_a$ labels the conjugate forces, moments or stresses. For small perturbations about an undeformed state, we may write $\sigma_a = M_{ab}u_b$. A generic medium is said to obey Maxwell-Betti reciprocity if and only if $M_{ab}$ is symmetric, i.e. $M_{ab} = M_{ba}$. As long as the system's linear response obeys Maxwell-Betti reciprocity, the forces can be derived from gradients of an energetic potential $V = \frac{1}{2}M_{ab}u_a u_b$. However, if the linear response of the medium violates Maxwell-Betti reciprocity, the internal energy is no longer a function of the coordinates $u_a$. In other words, the medium can perform nonzero work along a closed cycle of deformations[27]. Such a medium necessarily contains non-conservative forces that require an internal or external source of energy to be present.

While Maxwell-Betti reciprocity and momentum conservation are distinct, one can certainly design and build mechanical systems that harness linear or angular momentum sources to achieve violations of Maxwell-Betti reciprocity[7,20,29]. Such approaches, however, inherently involve mechanical coupling to an external medium. Here we take an alternative route. We explore the case where $\sigma_a$ represents shear and moment stresses, which conserve currents of linear and angular momentum. In this case $u_a$ represents geometric deformations and $M_{ab}$ represents the stiffness matrix containing all of the material's elastic moduli. We refer to the antisymmetric components $(M_{ab} - M_{ba})/2$ as *odd elasticity*[27,30–32]. A material displaying odd elasticity must violate Maxwell-Betti reciprocity, though it needs not rely on external sources of linear or angular momentum.

Recent advances in metamaterial design and prototyping have utilized active components to achieve functionalities such as sensing, lasing, and cloaking[10–12,15], frequency dependent reflectivity[33], unidirectional wave amplification[7,34,35], energy harvesting[16], and analog computation[17]. Nonetheless, all the active non-reciprocal metamaterials so-far realized exhibit either of the following fundamental limitations: the active non-reciprocal effects either vanish from the linear response in the quasistatic limit[33] or they require the presence of background sources of linear or angular momentum[7,29,36]. As a result, their functionalities are largely restricted to finite-frequency control or fundamentally require the sample to be in contact with an additional medium that acts as a momentum sink or source.

Here, we report the design, construction, and experimental demonstration of a freestanding metamaterial whose elasticity is unattainable in passive media (Fig. 1a–d). The metamaterial is constructed with piezoelectric elements[37–46] mounted on a beam and controlled by electrical circuits. Our approach enables an asymmetric coupling between bending and shearing in micropolar solids[47,48]. This results in an odd micropolar material that

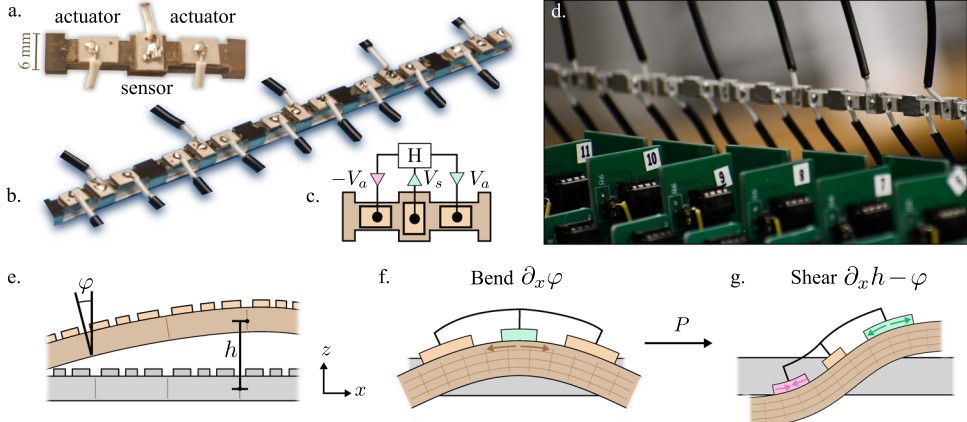

**Fig. 1 Design and mechanics of an odd micropolar metabeam. a** A single unit cell featuring three piezoelectric patches mounted on a beam: one that acts as a sensor, and two that act as actuators. **b** A segment of the full metabeam. **c** Each unit cell has an electronic loop. The voltage $V_s$ induced by the central piezoelectric is fed into a transfer function $H(\omega) = V_a(\omega)/V_s(\omega)$ that sends opposing voltages $V_a$ and $-V_a$ to the piezoelectric actuators. **d** A photograph of the metabeam (horizontal) with the electronic circuits in the foreground. We note that the mechanical forces from the attached wires are negligible. The wires act only as sources of energy and computation, but not of linear or angular momentum. **e** The motion of the metabeam can be described by two independent fields, $\varphi$ and $h$, which parameterize the angular and vertical displacements of the metabeam. Notice that under a reflection about the z-axis, we have $\varphi \to -\varphi$ and $h \to h$. **f** When the beam bends, the center piezoelectric is stretched. **g** The antisymmetric electronic actuation then gives rise to a shearing stress proportional to the modulus $P$.

simultaneously breaks parity and Maxwell-Betti reciprocity. The spectrum of the metabeam exhibits a non-Hermitian topological index which results in the localization of vibrational modes at sample boundaries. We experimentally show the resulting unidirectional amplification/attenuation of waves propagating through the metamaterial. Our work sheds light on controlling non-reciprocal elasticity in artificial materials.

## Results

**Design of the active metamaterial**. The metamaterial we design takes the form of a thick beam, whose shape is characterized by two independent degrees of freedom (Fig. 1e): the height $h(x)$ of the midplane and the angle $\varphi(x)$ of the cross section with respect to the vertical. A single unit cell in the beam is equipped with three piezoelectric patches that enable shape-sensing and response. The central patch acts as a sensor that acquires a voltage proportional to the elongation or contraction of the top surface (Fig. 1f). The piezoelectric patches at the front and back of the unit cell serve as mechanical actuators that elongate and contract in response to an applied voltage[49,50]. A transfer function $H(\omega)$ processes the input voltage from the central patch and sends output signals to the actuating patches (Fig. 1c). For each unit cell, we implement $H(\omega)$ using a minimal electrical circuit (Fig. 1d). The electronics only couple piezoelectric patches within a single unit cell, thereby creating a control system that is both local and decentralized. The resulting system only needs to be connected to a voltage or, more generally, energy source. One practical advantage of this approach is that the voltage or power source can easily be housed inside the medium itself.

The active metamaterial we design is freestanding—it does not push or pull on an external medium. Indeed, it obeys Newton's third law by preserving both angular and linear momentum as a traditional beam. However, the crucial difference between a traditional beam and the one we construct is the presence of internal energy sources. We design the feedback such that when the central patch experiences elongation or compression due to bending of the beam, the electronic loop produces output voltages that are antisymmetric (Fig. 1c), resulting in shear stresses (Fig. 1g). However, the ensuring shear strain does not stretch or compress the central piezoelectric patch. Therefore, the electromechanical control loop is entirely feed-forward: bending induces shear, while shear does not induce bending.

**Odd micropolar elasticity**. The feedback results in an elastic response that cannot be realized without an internal source of energy. This effect, apparent in the emergent continuum equations, may be deduced solely using symmetries and conservation laws based on classical beam theory (1D micropolar elasticity). Crucial to our design is the notion of a parity inversion P, defined here to be a mirror reflection of the beam about the $z$-axis that sends $x$ to $-x$. Fig. 1e shows that under parity, the two independent degrees of freedom, $h(x)$ and $\varphi(x)$, transform as

$$h(x) \underset{\rightarrow}{\mathcal{P}} h(-x) \tag{1}$$

$$\varphi(x) \underset{\rightarrow}{\mathcal{P}} -\varphi(-x) \tag{2}$$

Since $\varphi(x)$ acquires a minus sign under parity, we say that $\varphi(x)$ is a micropolar degree of freedom[47,48]. The equations of motion for a freestanding micropolar beam are then built out of conservation laws. Linear momentum conservation implies that

$$\rho \ddot{h} = \partial_x \sigma_{zx} \tag{3}$$

where $\sigma_{zx}$ is the shear stress and $\rho$ is the mass density. Moreover,

angular momentum conservation implies

$$I \ddot{\varphi} = \partial_x M + \sigma_{zx} \tag{4}$$

where $M$ is the bending moment and $I$ is the cross-sectional moment of inertia. The moment $M$ and stress $\sigma_{zx}$ are themselves determined by the deformation of the beam via a set of constitutive relations. To leading order in gradients of $h$ and $\varphi$, the internal geometry of the beam is approximated by two independent types of deformation: bending $b(x)$ (Fig. 1f) and shearing $s(x)$ (Fig. 1g), defined as

$$b(x) = \partial_x \varphi \tag{5}$$

$$s(x) = \partial_x h - \varphi \tag{6}$$

Under parity inversions, Eqs. (1, 2) imply that

$$b(x) \underset{\rightarrow}{\mathcal{P}} b(-x) \tag{7}$$

$$s(x) \underset{\rightarrow}{\mathcal{P}} -s(-x) \tag{8}$$

Assuming a linear response, one may in general write the linear constitutive relations as:

$$\begin{bmatrix} \sigma_{zx}(t) \\ M(t) \end{bmatrix} = \int_{-\infty}^{\infty} \begin{bmatrix} C_{11}(t') & C_{12}(t') \\ C_{21}(t') & C_{22}(t') \end{bmatrix} \begin{bmatrix} s(t-t') \\ b(t-t') \end{bmatrix} dt' \tag{9}$$

or in terms of frequency:

$$\begin{bmatrix} \sigma_{zx}(\omega) \\ M(\omega) \end{bmatrix} = \begin{bmatrix} C_{11}(\omega) & C_{12}(\omega) \\ C_{21}(\omega) & C_{22}(\omega) \end{bmatrix} \begin{bmatrix} s(\omega) \\ b(\omega) \end{bmatrix} \tag{10}$$

We denote the matrix operator on the right-hand side of Eq. (9) as $\mathbf{C}(t)$. Causality implies that $\mathbf{C}(t) = 0$ for $t < 0$ and reality of $\mathbf{C}(t)$ implies $\mathbf{C}(-\omega) = \mathbf{C}^*(\omega)$. Direct substitution of Eq. (10) into Eqs. (3, 4) reveals that the equations of motion are invariant under parity if and only if $C_{21}$ and $C_{12}$ are zero. Hence, we say that $C_{21}$ and $C_{12}$ are micropolar moduli.

It is useful to parameterize the $\mathbf{C}(\omega)$ as

$$\begin{bmatrix} C_{11}(\omega) & C_{12}(\omega) \\ C_{21}(\omega) & C_{22}(\omega) \end{bmatrix} = \begin{bmatrix} \mu(\omega) & \alpha(\omega) + \beta(\omega) \\ \alpha(\omega) - \beta(\omega) & B(\omega) \end{bmatrix} \tag{11}$$

where $\mu$ is the shear modulus and $B$ is the bending modulus, and $\alpha$ and $\beta$ are the symmetric and antisymmetric components of the micropolar moduli. If the beam lacks an internal source of energy, then the total work done by the beam on any deformation process that begins and ends in the same configuration must be less than zero: $\Delta W \leq 0$. This energy condition places the following constraints on the finite frequency linear response coefficients (see "Methods"):

$$0 \geq \text{Im}[\mu(\omega) + B(\omega)] \tag{12}$$

$$0 \leq \text{Im}[\mu(\omega)]\text{Im}[B(\omega)] - \text{Im}[\alpha(\omega)]^2 - \text{Re}[\beta(\omega)]^2 \tag{13}$$

for all $\omega$. Notice that the reality condition $\mathbf{C}(-\omega) = \mathbf{C}^*(\omega)$ implies $\text{Im}[\mathbf{C}] = 0$ at $\omega = 0$. As a consequence, we must have $\beta(\omega \rightarrow 0) = 0$ for any passive beams.

However, our active metamaterial has an internal source of energy and thus need not obey this constraint. The feed-forward coupling between bending and shearing suggests a linear response matrix of the form:

$$\begin{bmatrix} C_{11}(\omega) & C_{12}(\omega) \\ C_{21}(\omega) & C_{22}(\omega) \end{bmatrix} = \begin{bmatrix} \mu(\omega) & P(\omega) \\ 0 & B(\omega) \end{bmatrix} \tag{14}$$

The electronic control loop introduces the coefficient $P = 2\alpha = 2\beta$, which we refer to as the odd micropolar modulus. This modulus breaks two crucial symmetries. Since the electromechanical coupling violates parity, the active modulus $P$ must occur in

the off-diagonal entries. Moreover, $P$ occurs only in the upper-right entry because the electro-mechanical coupling is feed-forward: bend causes shear, but shear does not cause bend. As a result, the matrix **C** is asymmetric, indicating that the beam violates Maxwell-Betti reciprocity, even at zero frequency.

For our metabeam, we measure the moduli via COMSOL simulations in which we apply controlled displacements at finite frequency to the front and back faces of a single unit cell. By measuring the reaction forces on these faces, we determine the resulting stresses and, consequently, the moduli. We empirically find that $\mu = 1.3 \times 10^9\,\mathrm{kg/ms^2}$ and $B = 0.112 \times 10^6\,\mathrm{kg/s^2}$ are approximately independent of frequency, while $P(\omega) = \Pi H(\omega)$, where $H(\omega)$ is the transfer function and $\Pi = 4.7 \times 10^6\,\mathrm{kgm/s^2}$ is a material constant. From the metabeam geometry and materials, we compute the average volumetric density $\rho = 5613\,\mathrm{kg/m^3}$ and the average cross-sectional moment of inertia $5.9 \times 10^{-3}\,\mathrm{kg/m}$. See Supplementary Notes 1 and 2 for additional characterization details.

**Energy cycles**. It is useful to consider the elastic limit of Eq. (11) in which $\mathbf{C}(\omega)$ is real and approximately independent of frequency. For a passive beam in this limit, the matrix **C** is obtained by approximating the energetic cost of deformation by a quadratic function:

$$W = \frac{\mu}{2}s^2 + \frac{B}{2}b^2 + \alpha sb \qquad (15)$$

The work (per unit volume) done in an infinitesimal deformation of the beam is given by:

$$dW = \sigma_{zx}ds + Mdb \qquad (16)$$

From Eq. (16), we conclude $\sigma_{zx} = \frac{\partial W}{\partial s}$ and $M = \frac{\partial W}{\partial b}$. Hence, we obtain the linear constitutive relations:

$$\begin{bmatrix} \sigma_{zx} \\ M \end{bmatrix} = \begin{bmatrix} \mu & \alpha \\ \alpha & B \end{bmatrix} \begin{bmatrix} s \\ b \end{bmatrix} \qquad (17)$$

Since **C** in Eq. (17) is obtained via a second derivative of an energy function, it is symmetric. This is a manifestation of the Maxwell-Betti reciprocity theorem. In this case, the cumulative work done over a sequence of deformations depends only on the initial and final configurations: $\Delta W = \int dW = W_{\mathrm{final}} - W_{\mathrm{initial}}$. In particular, if the procedure begins and ends at the same state, we have $\Delta W = 0$.

Notice that the constitutive relation in Eq. (14) violates $\mathbf{C} = \mathbf{C}^{\mathrm{T}}$. Hence, the Maxwell-Betti reciprocity theorem implies that the form of **C** in Eq. (14) does not follow from a potential energy function. To see this explicitly, consider the differential of energy given by:

$$dW = \sigma_{zx}ds + Mdb \qquad (18)$$

$$= d\left(\frac{\mu}{2}s^2 + \frac{B}{2}b^2 + \frac{P}{2}sb\right) + \frac{P}{2}(bds - sdb) \qquad (19)$$

Notice that the second term in Eq. (19) cannot be expressed as the differential of a potential. By Green's theorem, integrating the work done $\oint dW$ over a closed loop in strain space yields:

$$\oint_{\partial V} dW = \oint_V Pdsdb = P \times \mathrm{Area} \qquad (20)$$

where "Area" is the signed area of the region $V$ enclosed by the path in the space of strains. Notice that the work done per cycle can be either positive or negative depending on the orientation of the path, and is independent of the rate of the process (in the quasistatic limit). Since the crucial ingredient for such cycles is an

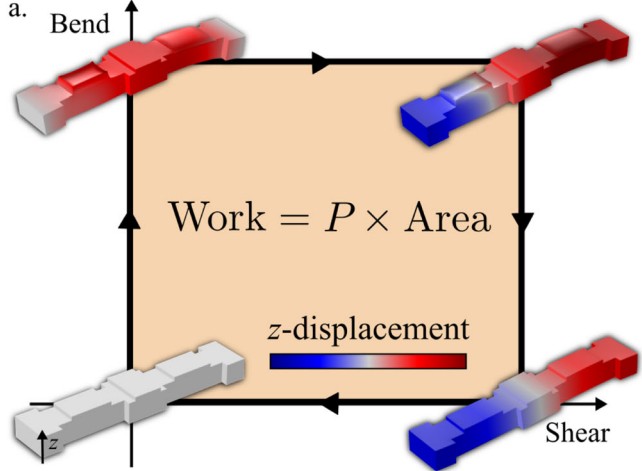

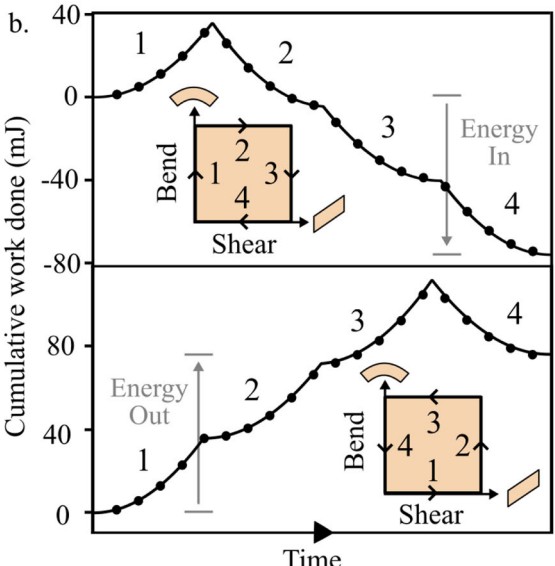

**Fig. 2 Quasistatic deformation cycles with odd micropolar elasticity.**
**a** The state of the unit cell is tracked in the space of shear and bend. When a quasistatic closed path is traced out in this space, the unit cell performs work per unit volume that is proportional to the area times the modulus $P$. The $z$-displacement is provided in arbitrary units. **b** We numerically compute the work done for a clockwise (top) and a counterclockwise (bottom) path. The solid lines are predictions from the continuum theory, and the black dots result from finite element simulations of the unit cell. In the simulations, maximum amplitudes of bending and shearing are $b_{\mathrm{max}} \approx 10^{-1}\,\mathrm{m^{-1}}$ and $s_{\mathrm{max}} \approx 10^{-2}$, respectively. See Supplementary Note 1 for further details on the simulation.

antisymmetry in **C**, we refer to this form of elasticity as "odd" (i.e., antisymmetric) elasticity [27].

To verify that our design displays this property at low frequencies, we perform COMSOL simulations of the beam with full piezoelectric coupling. As illustrated in Fig. 2a, we subject a single unit cell to a four-step protocol of shearing and bending by enforcing displacement-controlled boundary conditions at the two ends of the beam. We measure the reaction forces on the control surfaces to compute the work done by the beam, plotted in Fig. 2b. When the deformations are performed in a clockwise direction in strain space, as shown in the top panel of Fig. 2b, the cumulative work done is negative once the unit cell returns to its

initial configuration. Hence, energy flows from the external agent into the internal power reserves of the medium. When the cycle is reversed, so is the flow of energy. The ability to inject or extract mechanical energy through quasistatic cycles is synonymous with odd elasticity, i.e. a quasistatic stress-strain relationship that does not follow from a potential energy. In the quasistatic limit, the total work done is given by $P$ times the area enclosed in strain space. In the Methods, we also derive energy relations for cycles at finite frequency. For those cases, the total work done is related to $|P|$, $\arg(P)$, and the area enclosed in strain space.

**Odd micropolar elastodynamics and the non-Hermitian skin effect**. We now ask about the dynamic consequences of the active feed-forward control with parity violation. We first consider the elastic approximation in which $\mathbf{C}(\omega)$ is real and frequency independent. The linearized continuum equations governing the motion of the beam are given by:

$$\rho\ddot{h} = \mu\partial_x^2 h + P\partial_x^2\varphi - \mu\partial_x\varphi \tag{21}$$

$$I\ddot{\varphi} = \mu\partial_x h + B\partial_x^2\varphi + P\partial_x\varphi - \mu\varphi \tag{22}$$

Using Fourier transform, Eqs. (21, 22) may be cast in the form:

$$\omega\begin{bmatrix}\tilde{p}_h\\\tilde{p}_\varphi\\\tilde{s}\\\tilde{b}\end{bmatrix} = \omega_1\underbrace{\begin{bmatrix}0 & 0 & -kl_1 & -\tilde{P}l_1k\\0 & 0 & i & i\tilde{P}-kl_2\\-kl_1 & -i & 0 & 0\\0 & -kl_2 & 0 & 0\end{bmatrix}}_{D(k)}\begin{bmatrix}\tilde{p}_h\\\tilde{p}_\varphi\\\tilde{s}\\\tilde{b}\end{bmatrix} \tag{23}$$

where $k$ and $\omega$ are the wave number and frequency associated with the Bloch wave $e^{i(kx-\omega t)}$. We have introduced the notation:

$$\tilde{s} = \sqrt{\mu}(\partial_x h - \varphi), \quad \tilde{b} = \sqrt{B}\partial_x\varphi \tag{24}$$

$$\tilde{p}_\varphi = \sqrt{I}\dot{\varphi}, \quad \tilde{p}_h = \sqrt{\rho}\dot{h} \tag{25}$$

Here, $\tilde{s}$ represents the shear, $\tilde{b}$ represents the bending, $\tilde{p}_\varphi$ is the angular momentum, and $\tilde{p}_h$ is the $z$-component of the linear momentum. This parameterization is natural since the standard inner product

$$2e = \left|\tilde{p}_h\right|^2 + \left|\tilde{p}_\varphi\right|^2 + |\tilde{s}|^2 + \left|\tilde{b}\right|^2 \tag{26}$$

is equal to twice the mechanical energy density $e$. The dynamical matrix $D(k)$ depends on four parameters: $l_1 \equiv \sqrt{I/\rho}$ is roughly the thickness of the metabeam; $l_2 \equiv \sqrt{B/\mu}$ is the distance over which shearing and bending of equal transverse deflection cost equal amounts of energy; $\omega_1 \equiv \sqrt{\mu/I}$ sets a frequency scale separating transverse flexural modes and high frequency shearing modes; finally, the parameter $\tilde{P} \equiv P/\sqrt{B\mu}$ is the normalized odd micropolar modulus. For our metabeam $l_1 \approx 10^{-3}$ m, $l_2 \approx 10^{-2}$ m, $\omega_1 \approx 10^5$ Hz, and $\tilde{P} \lesssim 1$.

Within the continuum theory, the vibrational dynamics can be captured by solving the secular equation $\det D(k)[-\omega] = 0$, which takes the form

$$0 = \tilde{\omega}^4 - \left[1 - i\tilde{P}kl_2 + k^2\left(l_1^2 + l_2^2\right)\right]\tilde{\omega}^2 + k^4 l_1^2 l_2^2 \tag{27}$$

where $\tilde{\omega} = \omega/\omega_1$. Notice that Eqs. (21, 22) are two coupled second order equations and hence permit a dispersion with four branches. For small wavenumber, the dispersion for the low frequency flexural bands is given by:

$$\omega_\pm = \pm\omega_1\left[l_1 l_2 k^2 \pm i\tilde{P}l_1 l_2^2 k^3 + \mathcal{O}\left(l_1^2 l_2^2 k^4\right)\right] \tag{28}$$

As can be seen from Eq. (28), when $P$ is nonzero and real, the periodic boundary spectrum acquires a nonzero imaginary

component. The nonzero imaginary contribution arises since the active metabeam has the ability to physically introduce or remove mechanical energy. Moreover, the modulus $P$ violates parity and hence breaks the symmetry between $k \to -k$, in contrast to the design in ref. [7] where the parity violation is induced by a term in the dispersion that is linear in $k$.

The simultaneous breaking of parity and energy conservation allows our active micropolar metamaterial to selectively amplify and attenuate waves based on their direction of travel. In Fig. 3a, we show the spectrum of the flexural mode in the right half of the complex plane for $P > 0$. The solid line is the continuum theory, valid at small $k \ll 1/\sqrt{l_1 l_2}$, and the discrete points are the results of fully coupled COMSOL simulations. In the calculations, $P = \pm 3\Pi$, and there are no free fitting parameters. (The material constants $\Pi$, $\rho$, $B$, and $\mu$ in the continuum theory are determined from independent simulations, see Odd micropolar elasticity.) As illustrated in Fig. 3a, for $P > 0$, $\mathrm{Im}(\omega) > 0$ whenever $\mathrm{Re}(d\omega/dk) > 0$ and $\mathrm{Im}(\omega) < 0$ whenever $\mathrm{Re}(d\omega/dk) < 0$. Physically, this means that wave packets traveling to the right are amplified, while wave packets traveling to the left are attenuated. As an illustration, Fig. 3b shows the inverse penetration depth $\kappa$ for $\omega$ taken to lie along the positive real axis. From the continuum theory, we compute the analytical formula for $\kappa$ (see "Methods")

$$\kappa = \sqrt{\frac{\rho}{B}}\frac{P}{4\mu}\omega \tag{29}$$

In Fig. 3b, we compare Eq. (29) to the COMSOL simulations (see "Methods") and a semi-analytical technique known as the transfer matrix method (see Supplementary Note 3).

This unidirectional amplification can be understood from the point of view of the non-Hermitian skin effect[51–62]. Given a complex frequency $\omega$, we can define the following topological index:

$$\nu(\omega) = \frac{1}{2\pi i}\sum_\alpha \oint_{-\pi/L}^{\pi/L}\frac{\mathrm{d}}{\mathrm{d}k}\log\left[\omega_\alpha(k) - \omega\right]\mathrm{d}k \tag{30}$$

where $L$ is the length of a single unit cell and $\omega_\alpha(k)$ is the frequency of the $\alpha$ band. Here, we take $\alpha$ to run over the flexural bands. The topological index $\nu(\omega)$ indicates whether a system with semi-infinite boundary conditions will host a localized mode at the frequency $\omega$. When $\nu(\omega) > 0$, a semi-infinite system with domain $x \in [0,\infty)$ will host a mode localized to its left boundary at frequency $\omega$. Likewise, when $\nu(\omega) < 0$, a semi-infinite system with domain $x \in (-\infty,0]$, will host a mode localized to its right boundary.

As shown in Fig. 3c, d, the sign of $\kappa$ can be rationalized by examining $\nu(\omega)$ for an example frequency $\omega$ (denoted by the star) along the real axis. For $P > 0$, the periodic boundary spectrum (red) winds once counter-clockwise around the star, and hence $\nu(\omega) < -1$. However, for $P < 0$, the localization is reversed since the direction of the contour is reversed. In the Methods, we show how to compute $\nu(\omega)$ directly from the continuum equations using a generalization of Eq. (30). Furthermore, in Supplementary Note 1 we discuss how the presence of additional vibrational bands affect the calculation and physical interpretation of $\nu(\omega)$.

**Pseudo-Hermitian dynamics and direction-dependent bending modulus**. In the preceding section, we took $P(\omega)$ to be real and constant. However, by introducing a phase delay into the transfer function $H(\omega)$ we can control the complex argument of $P$. When $\arg(P) = \pm\pi/2$, the secular Eq. (27) has entirely real coefficients. Hence, the periodic boundary spectrum will consist of frequencies that come in real values or complex conjugate pairs. This

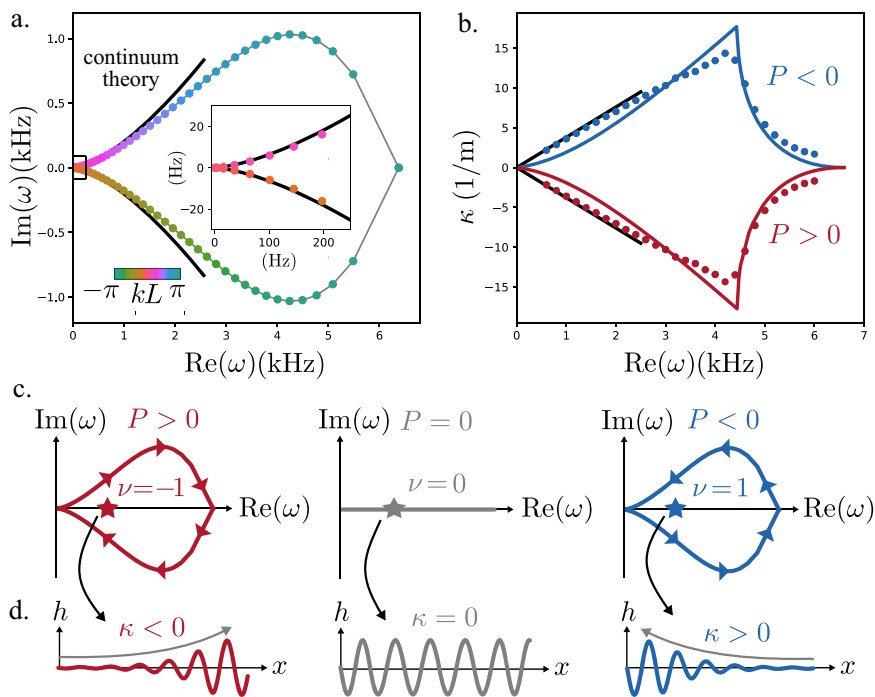

**Fig. 3 Non-Hermitian skin effect via the odd micropolar elasticity. a** The vibrational spectrum for the flexural mode of a metabeam with periodic boundary conditions and odd micropolar modulus $P = 3\Pi$. The black line results from the continuum theory given by Eq. (28). The data points are obtained via fully piezoelectrically coupled simulations in COMSOL with the hue indicating the wavenumber $kL$, where $L$ is the unit cell length. For the full spectrum plotted as a function of $k$ in the continuum theory and in the numerics, see Fig. 7 and S2, respectively. The inset compares the continuum theory and simulations for small wavenumbers $\ll 1/\sqrt{l_1 l_2}$. **b** The inverse penetration depth $\kappa$ for real $\omega$ in a medium with open boundary conditions. The points are the results of COMSOL simulations, the black lines are Eq. (29), and the dark lines are the result of the transfer matrix method, see Supplementary Note 3. **c** The localized states are connected to a topological index $\nu(\omega)$. The periodic boundary spectrum for $P > 0$, $P = 0$, and $P < 0$ are represented schematically by the solid lines. The arrows indicate the direction of increasing $k$. For a given frequency $\omega$, the winding number $\nu(\omega)$ of the periodic boundary spectrum indicates the presence of a localized mode. **d** The localization of eigenmode at the value of $\omega$ denoted by the star in (**c**) is schematically illustrated.

additional symmetry is sometimes referred to as a generalized PT symmetry[30,63,64], which arises if and only if there exists an antiunitary operator that commutes with $\boldsymbol{D}(k)$. We say that the PT symmetry is unbroken when the eigenvalues of $\boldsymbol{D}(k)$ are entirely real, and that it is broken otherwise. In the unbroken phase, $\boldsymbol{D}(k)$ is said to be pseudo-Hermitian. Pseudo-Hermiticity implies that each eigenvector of $\boldsymbol{D}(k)$ will individually conserve the mechanical energy density $e$ in Eq. (26). However, when two or more eigenvectors are superimposed, $e$ can oscillate in time, though remaining centered around a constant time-averaged value.

Since pseudo-Hermiticity constrains the periodic boundary spectrum of $\boldsymbol{D}(k)$ to lie along the real line, the unidirectional amplification vanishes when $P$ is imaginary. Nonetheless, the effects of parity violation are still present. We note that Eq. (28) may be written in the form:

$$\omega_{\pm} = \pm\sqrt{\frac{B}{\rho}\left(1 \pm i\frac{Pk}{\mu}\right)k^2 + \mathcal{O}\left(k^4\frac{I^2 B^2}{\rho^2\mu^2}\right)} \quad (31)$$

When $\arg(P) = \pm\pi/2$, we can interpret the form of Eq. (28) as having a rescaled bending modulus:

$$B_{\text{eff}} = B\left(1 \mp \frac{|P|k}{2\mu}\right)^2 \quad (32)$$

Notice that the value of $B_{\text{eff}}$ depends on the sign of $k$, and hence the effective bending modulus is direction dependent. Fig. 4a shows the tilting of the dispersion for $\arg(P) = \pm\pi/2$. The tilt

implies that the phase and group velocities for right and left traveling waves are unequal. Numerically solved modes are shown in Fig. 4b. We note that the pseudo-Hermiticity endowed by $P(\omega) \propto \pm i$ must exist exclusively at finite frequency because $P(\omega)$ cannot be nonzero and imaginary at $\omega = 0$ due to the requirement that $P(-\omega) = P^*(\omega)$.

**A discrete model of the odd micropolar metabeam.** To gain intuition into the mechanics of the metabeam, it is useful to consider a discrete model. As shown in Fig. 5a, b, the $i$th unit cell of the discrete model consists of a rod (gray) with moment of inertia $J$ and total mass $m$, whose position and orientation are captured by a height $h_i$ and an angle $\varphi_{i-1}$. The rods are connected via mass-less, rigid frames and two springs. The top spring is a Hookean spring with tension

$$T_i = k_\mu(h_{i-1} - h_i + L\varphi_{i-1}) \quad (33)$$

and a bottom spring is a torsional spring that exerts an angular tension

$$\tau_i = \kappa_B(\varphi_{i-1} - \varphi_i) \quad (34)$$

In Fig. 5c, d, we show the addition of an active element that senses the stretching of the bottom spring and actuates an additional tension in the top spring

$$T_i^a = p(\varphi_{i-1} - \varphi_i) \quad (35)$$

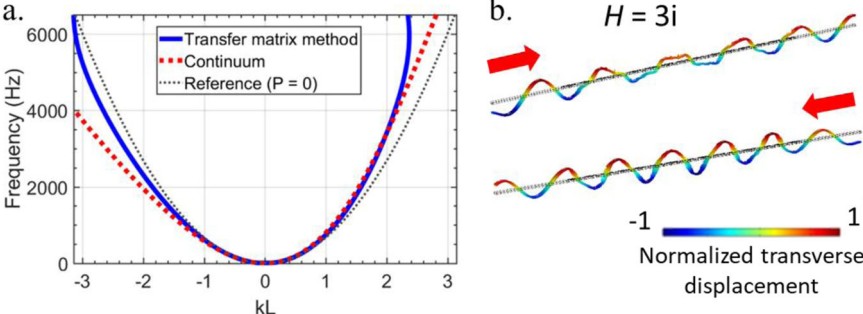

**Fig. 4 Pseudo-Hermitian dynamics. a** The spectrum is shown for the metamaterial with $\arg(P) = \pi/2$. We note that the reality of the frequencies is maintained, while the modulus $P$ breaks the $k \to -k$ symmetry. $L$ is the unit cell length. **b** Transverse displacement wave fields for the waves traveling in different directions. The left and right traveling modes are excited at equal frequencies, but have differing wavenumbers due to the odd micropolarity. The red arrows indicate the direction of travel of the wave, and $H$ is the transfer function such that $P = \Pi H$.

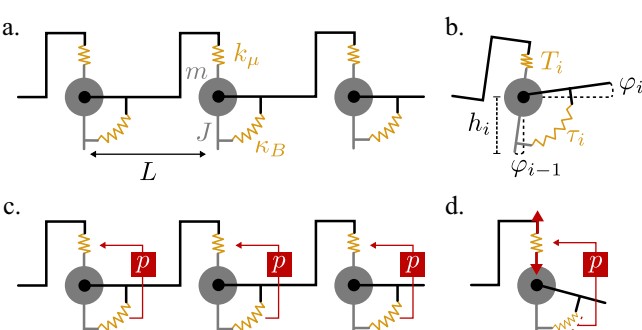

**Fig. 5 Discrete model for odd micropolar beam. a** A discrete model of a Timoeshenko beam consists of a central mass $m$ with moment of inertial $J$, a Hookean spring of spring constant $k_\mu$ and a torsional spring of spring constant $\kappa$, Band lattice spacing L. **b** The unit cell is described by the height of the mass $h_i$ and the angle $\varphi_i$ of the black connecting rod. **c**, **d** The odd micropolar beam has an internal feedback $P$ that senses the angle change of the torsional spring and actuates additional tension in the Hookean spring. The control loop is unidirectional: stretching or compressing the Hookean spring does not affect the torsional spring.

Summing the forces in the vertical direction yields a dynamical equation for $h_i$

$$m\ddot{h}_i = T_i - T_{i+1} + T_i^a - T_{i+1}^a \tag{36}$$

$$= k_\mu(h_{i+1} + h_{i-1} - 2h_i) + Lk_\mu(\varphi_{i-1} - \varphi_i) \\ + p(\varphi_{i+1} + \varphi_{i-1} - 2\varphi_i) \tag{37}$$

Furthermore, summing the torques yields

$$J\ddot{\varphi}_i = \tau_i - \tau_{i+1} - L(T_{i+1} + T_{i+1}^a) \tag{38}$$

$$= \kappa_B(\varphi_{i+1} + \varphi_{i-1} - 2\varphi_i) + pL(\varphi_{i+1} - \varphi_i) \\ + Lk_\mu(h_{i+1} - h_i - L\varphi_i) \tag{39}$$

Upon inspection, Eqs. (37) and (39) are precise discretizations of Eqs. (21) and (22) with $\rho = m/L$, $I = J/L$, $\mu = k_\mu L$, $B = \kappa_B L$, and $P = Lp$. See Fig. S4 for a comparison of the dispersion for the discrete model and continuum theory. Notice that the discrete model manifestly conserves linear and angular momentum since the linear and torsional springs exert equal and opposite forces and torques, respectively, on the units they connect. Even without externally applied torques, nontrivial internal angular momentum transfer occurs between the translation of its center of mass $jLm\dot{h}_j$ and the rotation of the axis $J\dot{\varphi}_j$, akin to a "spin-orbit" coupling.

Nonetheless, Maxwell-Betti reciprocity is violated by the asymmetry in the relationship between the linear and torsional springs: bending of the torsional spring induces a tension $T_i^a = p(\varphi_{i-1} - \varphi_i)$ in the linear spring, while the deformation of the linear spring $h_{i-1} - h_i + L\varphi_{i-1}$ has no response in the angular spring. This asymmetry implies that a cycle of alternating actuation and release of the linear and torsional spring is associated with a nonzero amount of work done. For additional discrete models illustrating the independence of Maxwell-Betti reciprocity and momentum conservation, see Supplementary Note 1.

**Experimental demonstration.** To probe the dynamic wave phenomena originating from the odd micropolar modulus $P$, we perform experiments in which we excite flexural waves in the metabeam using piezoelectric actuators, see Fig. 6a and "Methods". In experiments, we implement the following electronic transfer function (cf. Fig. 1c):

$$H(\omega) = \frac{H_0}{(i\omega/\omega_0)^2 + 2i\zeta\omega/\omega_0 + 1} \tag{40}$$

Here, $\zeta = 0.48$ and $H_0 = 3$ are constants and $\omega_0 = 3$ kHz is the cutoff frequency at which $\arg(P) = -\pi/2$. Other electrical circuit and geometric parameters of the metabeam can be found in Supplementary Note 2. We probe the vibrational dynamics of the beam by initiating waves from either the right or left side of the metamaterial via external piezoelectric elements. Fig. 6b, c shows the experiment at 2 kHz in which waves from the right are suppressed while waves from the left are amplified (see also Supplementary Movies 1 and 2).

To construct the full spectrum of the metamaterial, we perform the experiment with tone burst signals centered between 1.5 and 4 kHz. The transverse velocity wave fields are measured along the medium using a laser Doppler vibrometer. We apply fast-Fourier transforms in time and in space to extract the real $k(\omega)$ and imaginary part $\kappa(\omega)$ of the wavenumber as a function of frequency for right-going (red) and left-going (blue) waves (Fig. 6d, e). As described in Supplementary Note 3, the solid theoretical curves are produced using a semi-analytical technique known as the transfer-matrix method. The transfer-matrix method utilizes the beam geometry, known material parameters, and electronic feedback measured in simulations. No fitting parameters are used in the comparison between experiment and the transfer-matrix method curve.

As illustrated in Fig. 6f, the transfer function $H(\omega)$ is chosen such that $-\pi/2 < \arg(P) < 0$ when $\omega < \omega_0$. In this case, we find

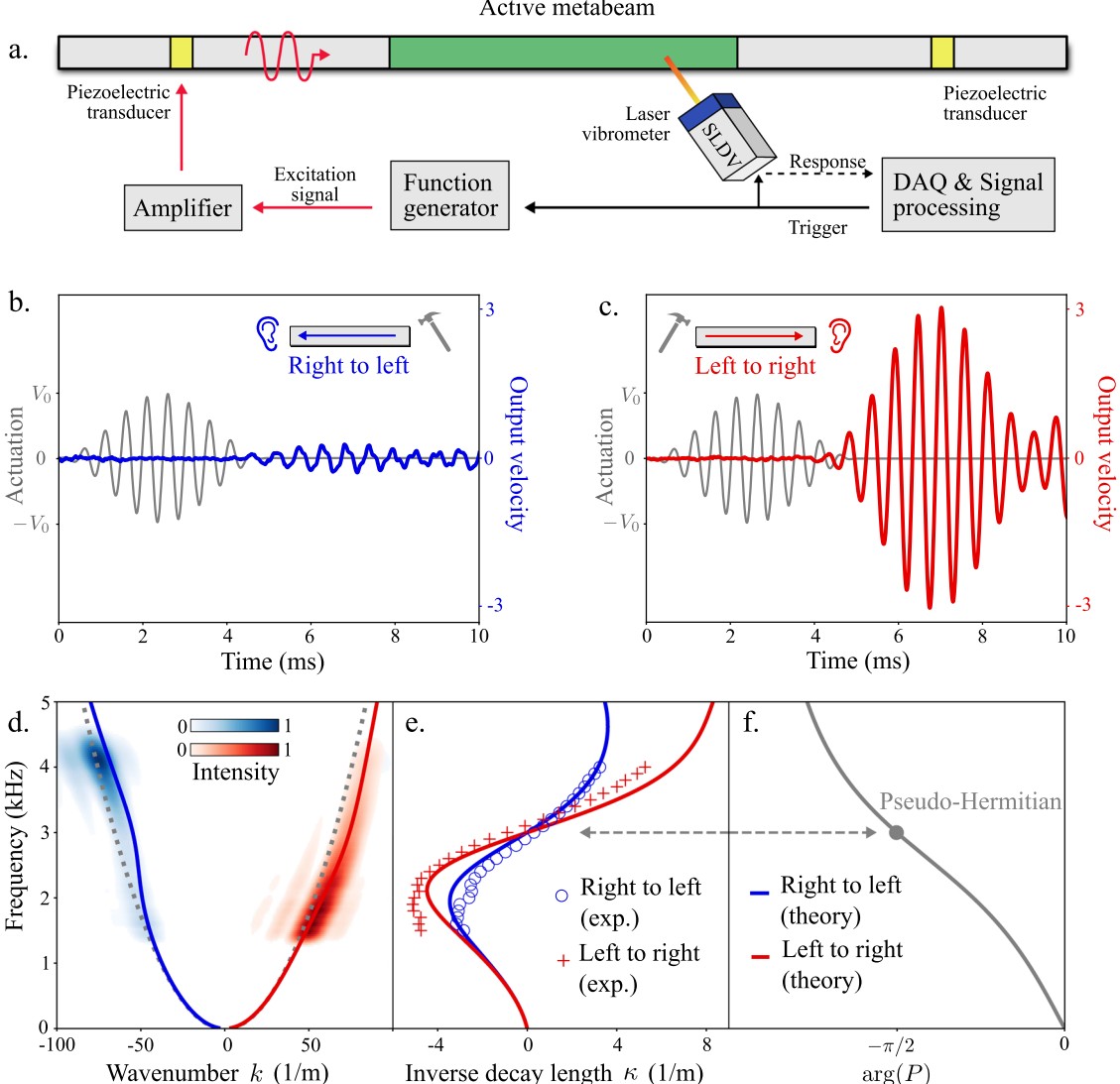

**Fig. 6 Experimental demonstration of skin modes and odd micropolar moduli. a** Experimental schematic. Flexural waves are generated in the active metabeam from either the right or left side using piezoelectric actuators (yellow), see "Methods". A scanning laser Doppler vibrometer (SLDV) measures the transverse velocity of the surface of the active metabeam. **b**, **c** Unidirectional amplification of waves. A metamaterial consisting of 9 unit cells is actuated from either the right (blue) or left (red) with a 2 kHz tone burst signal (gray). The output velocity is normalized by the maximum velocity observed when the experiment is performed with no active feedback. **d** Observation of the non-Hermitian skin effect. Experiments are performed between 1.5 kHz and 4 kHz for right to left (blue) and left to right (red) traveling waves. A 2D FFT shows the intensity of the observed spectrum. The intensity is normalized by its maximum value. **e** The inverse decay length. In **d**, **e** the solid theoretical curves are based on the transfer matrix method. In **d** the gray dashed curves are theoretical predictions with no activity. **f** A plot of $\arg(P)$ as a function of frequency. At $\omega = \omega_0 (= 3 \text{ kHz})$, $\arg(P) = -\pi/2$, indicating that the system is pseudo-Hermitian and accordingly we observe $\kappa = 0$ at $\omega = \omega_0$.

$\kappa < 0$ for both left- and right-propagating waves (panel e). A value of $\kappa < 0$ implies that waves propagating to the left are attenuated whereas waves propagating to the right are amplified, which is confirmed by the FFT intensity in panel d. Likewise $-\pi < \arg(P) < -\pi/2$ for $\omega > \omega_0$. In this case, we find $\kappa > 0$, indicating that waves propagating to the right are attenuated whereas waves propagating to the left are amplified. In addition, when $\omega = \omega_0$, $\arg(P) = -\pi/2$, and the waves propagating to the left and right display no attenuation or amplification. At this frequency, the effective dynamical matrix is pseudo-Hermitian, and the differences between left- and right-propagating waves reside only in the wavelengths and phase velocities. We note that, in experiments, the unidirectional amplification is limited by the maximum output voltage (±45 V) of our electric control system. In practice, to maximize the amplification ratio, one strategy is to

use a modest value of the transfer function, say $|H_0| \approx 3$, and increase the number of unit cells over which the wave is amplified.

## Discussion

The metabeam presented here demonstrates active odd micropolar moduli and non-reciprocal responses absent in energy conserving media that are enabled by sensing, actuating and local computation. The minimal on-board electronics that power the active metabeam enable its multiple functions as an elastic engine, selective mode amplifier, and mechanical diode. We uncover an intrinsic relation between an odd micropolar modulus, the non-Hermitian skin effect, and a corresponding topological index. Numerical and experimental results show unidirectional

amplification and attenuation of waves propagating through the metamaterial. Odd micropolarity extends the range of possible couplings between conventional strains/stresses and higher-order curvatures/moments by including antisymmetry in their relationship. The electronically assisted mechanical feedback provides an appealing solution to precisely modulate odd micropolar moduli without requiring changes to the metabeam's structure, geometry, or passive moduli. Our design can be flexibly tuned through computer coding and scaled via microelectromechancial systems (MEMS)[65,66]. Our mechanical approach relies on a feed-forward control loop, a generic concept that can exist in both metamaterial and biological contexts. The continuum theory also makes our approach especially generalizable to the mechanics in other systems. Combining the principles illustrated here with disorder, nonlinearities, and strong dissipation suggests new approaches for the control of filaments and membranes arising in biological media[4,18,19].

## Methods

**Sample fabrication**. The metabeam is composed of three piezoelectric patches (STEMiNC PZT 5J: 6 mm × 4 mm × 0.55 mm) mounted via conductive epoxy onto a laser-cut stainless steel host beam. We achieve antisymmetric actuation without the use of an inverting voltage amplifier by mounting the two piezoelectric actuators such that their piezoelectric polarization directions are oppositely oriented.

**Experimental procedures**. In experiments, nine metamaterial unit cells are connected with control circuits, see Fig. 6a. Two piezoelectric transducers are attached on the left and right sides of the metamaterial to generate incident flexural waves. We employ ten-peak tone-burst signals with central frequencies ranging from 1.5 to 4.0 kHz in step sizes of 0.1 kHz. We generate and amplify incident wave signals via an arbitrary waveform generator (Tektronix AFG3022C) and a high voltage amplifier (Krohn-Hite), respectively. Transverse velocity wavefields are measured on the surface of the metamaterial by a scanning laser Doppler vibrometer (Polytec PSV-400). We note that the transfer-matrix method used to derive the theoretical curves in Fig. 6d, e rigorously assumes an infinite system. To experimentally approximate these conditions, we embed the active metamaterial within a larger host steel beam denoted by the gray region of Fig. 6a. When waves cross the boundaries from host beam to the metamaterial, the reflection at boundaries between the host beam and the metamaterial is negligible, as evidenced by our numerical and experimental results. To suppress reflected waves at the free boundaries of the host beam, we bonded two layers of clay on the host beam with sufficient lengths. This way, waves can propagate through the metamaterial with approximated infinite boundary conditions. The decay length is then obtained by calculating the wave amplitudes at different points in the metamaterial.

**Finite element simulations**. We calibrate the transfer matrix method and continuum equations by conducting fully three-dimensional numerical simulations of the unit cell using the commercial finite element software COMSOL Multiphysics. In all the simulations, we model the piezoelectric patches via a three-dimensional linear piezoelectric constitutive law. The central piezoelectric patch acts as a sensor whose signal is obtained by integrating the free charge over the top surface of the piezoelectric sensor. The top and bottom surfaces of the piezoelectric sensor have zero electric potential. The bottom surfaces on the piezoelectric actuators are ground, and we apply electrical potentials on the top surfaces to act as actuating voltages. The actuating voltages are related to the sensing voltages via the electronic transfer function. For the wave dispersion computations in Fig. 3a, Floquet periodic boundary conditions are applied on the left and right boundaries of a metamaterial unit cell. We calculate eigenfrequencies of the unit cell with different real wavenumbers. To simulate the wave propagation with open boundaries (Fig. 3b), a metabeam composed of 15 unit cells is placed between two external beams. Two perfectly matched layers (PMLs) are attached to both ends of the external beams in order to suppress reflected waves from the boundaries. The incident flexural wave is generated by applying a harmonic force on the boundary of the host beam. The out-of-plane displacement is measured at the left- and right-hand sides of the metabeam. The penetration depth is calculated by comparing the amplitudes of the two extracted displacements.

**Energy bounds on dynamic moduli**. Here we discuss Eqs. (12, 13) in the main text. For simplicity, let us collect the stresses into a vector $\mathbf{t} = (\sigma_{zx}, M)^T$ and the deformations into a vector $\mathbf{u} = (s, b)^T$. Suppose the beam is subject to a deformation procedure such that its initial and final configurations at times $t = -\infty$ and $t = \infty$ are identical. Then the total work per unit volume done by the beam is given

by

$$\Delta W = \int_{-\infty}^{\infty} \frac{d\mathbf{u}}{dt} \cdot \mathbf{t} \, dt \tag{41}$$

$$= -i \int_{-\infty}^{\infty} \omega \mathbf{u}^{\dagger}(\omega) \cdot \mathbf{C}(\omega) \cdot \mathbf{u}(\omega) \, d\omega \tag{42}$$

$$= \int_{-\infty}^{\infty} \omega \mathbf{u}^{\dagger}(\omega) \cdot \mathbf{M}(\omega) \cdot \mathbf{u}(\omega) \, d\omega \tag{43}$$

In the final step, we have introduced the matrix $\mathbf{M}(\omega) = i[\mathbf{C}^{\dagger}(\omega) - \mathbf{C}(\omega)]$ and used the fact that $\mathbf{u}(-\omega) = \mathbf{u}^*(\omega)$ and $\mathbf{C}(-\omega) = \mathbf{C}^*(\omega)$. If the medium is passive, then we require that $\Delta W$ must be negative for all choices of $\mathbf{u}(\omega)$. Therefore, we require that the matrix $\mathbf{M}(\omega)$ be negative semidefinite. Using the parameterization in Eq. (11), this requirement implies Eqs. (12, 13). See refs. [67–69] for related discussions in two- and three-dimensional media.

**Cycles at finite frequency**. To gain intuition on the elastodynamics of the odd micropolar metabeam, it is useful to consider the notion of a cycle at finite frequency. At a finite frequency $\omega$, the modulus $P$ need not be real and we may write $P = |P| \, e^{i\varphi_P}$. In this case, both the real and imaginary parts of $P$ will contribute to the energy extracted. For example, consider a cyclic protocol that involves bending of amplitude $|b|$ and shearing of amplitude $|s|$ and relative phase delay $\varphi_B$. Applying Eq. (43), the total energy extracted per cycle is

$$\text{Work} = -\pi |P| |s| |b| \sin(\phi_B + \phi_P) \tag{44}$$

Notice that Eq. (44) gives mechanistic insight into the amplification of the waves observed in the experiment. When $P$ is nonzero, the eigenmodes of $\boldsymbol{D}(k)$ comprising a given plane wave will generically have a phase delay between bending and shearing. Hence, after one cycle, the nonconservative stresses will have converted stored electrical energy into mechanical energy. This conversion will cause the amplitude of the eigenvector to grow in proportion to its current amplitude, for which $|s| \, |b|$ is a proxy. Hence, the mode will be exponentially amplified, as reflected by the imaginary component of the eigenfrequencies.

**Topological index in the continuum**. In this section, we discuss the index $v(\omega)$ from the point of view of the continuum theory[54]. The spectrum as a function of $k$ is plotted in Fig. 7a for $P = 0$. The spectrum is given by the roots of the secular Eq. (27) and contains four roots since the equations of motion are second order in time and involve two coupled fields. We explicitly compute the eigenvectors and eigenvalues for small $k$ and $P = 0$ in Supplementary Note 1. The spectrum consists of a pair of Goldstone modes, which for small $k$ and $P = 0$ represents a flexural motion of the beam. Additionally, the continuum equations imply two modes separated by a band gap $\omega_1 \approx 105$ Hz. As shown in Supplementary Note 1, for small $k$ and $P = 0$, these modes are dominated by a shearing motion. For the Goldstone mode, we expect a range of validity of the continuum theory at small $k$, since $\omega(k \to 0) = 0$. However, for the shear dominated mode, the continuum theory is not expected to self-consistently apply due to the finite gap. In Supplementary Note 1, we numerically compute the spectrum and eigenmodes for frequencies above the experimentally relevant range using COMSOL. There we also discuss why the winding number $v(\omega)$ computed in the continuum is still physically relevant despite the presence of additional high frequency modes not captured by the continuum.

For a translationally invariant system, the difference between periodic and open boundary conditions is whether the differential operator that defines the equations of motion allows formal eigenvectors with complex wavenumber $k$. For a system with periodic boundary conditions, the spectrum consists of values of $\omega$ that solve $\det[\boldsymbol{D}(k) - \omega] = 0$ for real $k$. However, for a system on a semi-infinite domain $x \in [0, \infty)$, we allow $\text{Im}(k) > 0$. These modes decay to the right as $\exp[(-\text{Im}k + i\text{Re}k)x]$ and therefore maintain a finite $L^2$ norm, which represents the mechanical energy density. To determine whether a given frequency $\omega$ is in the semi-infinite boundary spectrum, we first must count the number of solutions to $\det[\boldsymbol{D}(k) - \omega] = 0$ with $\text{Im}\,k > 0$. We do so by applying Cauchy's argument principle from complex analysis:

$$\tilde{\nu}(\omega) = \lim_{R \to \infty} \frac{1}{2\pi i} \oint_{\Gamma(R)} \frac{d}{dk} \log \det[\boldsymbol{D}(k) - \omega] dk \tag{45}$$

where $\Gamma(R)$ is a counter clockwise curve in the complex plane given by $[-R, R]$ together with $Re^{i\varphi}$ for $\varphi \in [0, \pi]$. The winding number $\tilde{\nu}(\omega)$ itself does not directly determine the number of localized modes, since we must also consider the boundary conditions placed at $x = 0$. Suppose that $\gamma$ independent homogeneous boundary conditions are placed at $x = 0$. Then the number of left localized modes will generically be given by:

$$\text{left localized modes at } \omega = \tilde{\nu}(\omega) - \gamma \tag{46}$$

Likewise, for a system with boundary $(0, \infty]$, the number of right localized modes is given by

$$\text{right localized modes at } \omega = d - \tilde{\nu}(\omega) - \gamma \tag{47}$$

where $d$ is the degree of $\det[\boldsymbol{D}(k) - \omega]$ as a polynomial in $k$. Whenever the right-hand

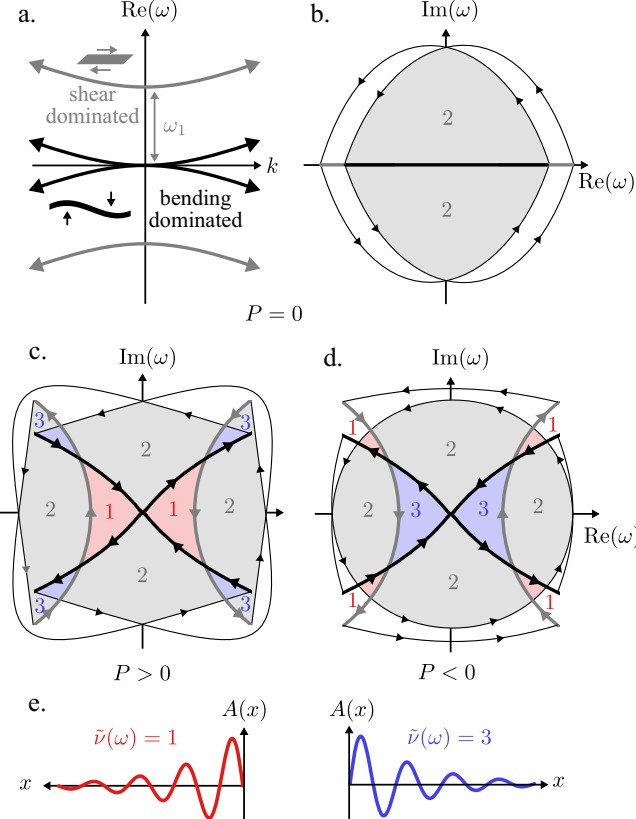

**Fig. 7 Non-Hermitian band topology via odd micropolar elasticity. a** The spectrum for $P = 0$ features a pair of bending dominated bands (black) and shear dominated bands (gray) separated by a band gap $\omega_1$. (b-d) The spectrum is shown in the complex $\omega$ plane for $P = 0$, $P > 0$, and $P < 0$. The thick black lines represent the bending dominated band, while the thick gray lines represent the shear dominated bands, both with $k \in [-R, R]$ for a finite $R$. The thin black lines represent the analytical continuation of the spectrum for $k = Re^{i\varphi}$ for $\varphi \in [0, \pi]$. The arrows indicate the direction of increasing $k$. The numbers indicate the value of $\nu(\omega)$ for $\omega$ in the corresponding colored regions of the complex plane. This number corresponds to the number of times that the spectrum winds around a given region. **e** For a semi-infinite system with a free boundary, the winding number of $\tilde{\nu} = 1$ ($\tilde{\nu} = 3$) for our continuum theory indicates a mode localized to the right (left) boundary. The wave forms schematically depict the localization with $A(x)$ representing amplitude. For a calculation of the precise eigenvectors, see Supplementary Note 1.

side of Eqs. (46) or (47) is negative, the mode count is taken to be zero. In Supplementary Note 1, we provide a derivation of Eqs. (46) and (47) and explicitly define independent, homogeneous boundary conditions. Such boundary conditions include, for example, stress-free ($M = \sigma_{zx} = 0$) or motion-free ($h = \varphi = 0$) boundaries. Fig. 7b–d illustrates the computation of the winding number. The thick black and gray lines are the periodic boundary spectrum for $k \in [-R, R]$, and the thin black lines are the analytical continuation for $k = Re^{i\varphi}$ for $\varphi \in [-\pi, \pi]$. Colored regions are labeled by the value of $\tilde{\nu}(\omega)$. Suppose for example that the beam is given stress and moment free boundary conditions $\sigma_{zx} = 0$ and $M = 0$ at $x = 0$. In this case $\gamma = 2$, and therefore $\tilde{\nu}(\omega) = 1$ indicates the presence of a right localized mode, $\tilde{\nu}(\omega) = 3$ indicates the presence of a left localized modes, as shown in Fig. 7e. In the Supplementary Note 1, we explicitly compute examples of the eigenmodes in Fig. 7c.

**Topological index from discrete models**. We now discuss the topological index in Eq. (30), which is appropriate for discrete settings such as the discrete model and finite element simulations. Suppose the system is composed out of a unit cell of finite length $L$ whose internal state is represented by a vector $\mathbf{\Psi}(x)$. The components of $\mathbf{\Psi}$ can represent, for example, the displacement and velocities of points in a finite element mesh. Here, $x$ is a discrete label that takes values in integer multiples of $L$. The Fourier transform of the equations of motion now read:

$$\omega \mathbf{\Psi}(k) = \mathscr{D}(k) \cdot \mathbf{\Psi}(k) \tag{48}$$

where $\boldsymbol{D}(k)$ is the dynamical matrix for the discrete system, and $k$ is the wave number assuming values in $[-\pi/L, \pi/L]$. For a system with periodic boundaries, the spectrum is given by solutions to $\det[\boldsymbol{D}(k) - \omega] = 0$ for $k \in [-\pi/L, \pi/L]$. For a system with semi-infinite boundaries (e.g., $x \in \{0, L, 2L, \ldots\}$), we allow $\text{Im} k > 0$. As detailed in the Supplementary Note 1, we can invoke a similar application of the Cauchy argument principle to determine the number of eigenmodes at a given frequency $\omega$. To do so, we write $k = -i\log\lambda$, where $\lambda$ assumes values on the unit circle $S$ in the complex plane. We can then apply Cauchy's argument principle using $S$ as a counterclockwise contour for $\lambda$. We have:

$$\nu(\omega) = \frac{1}{2\pi i} \int_S \frac{d}{d\lambda} \log \det\left[\mathscr{D}(-i\log\lambda) - \omega\right] d\lambda \tag{49}$$

$$= \frac{1}{2\pi i} \int_{-\frac{\pi}{L}}^{\frac{\pi}{L}} \frac{d}{dk} \log \det[\mathscr{D}(\mathbf{k}) - \omega] dk \tag{50}$$

$$= \frac{1}{2\pi i} \sum_\alpha \int_{-\pi/L}^{\pi/L} \frac{d}{dk} \log[\omega_\alpha(k) - \omega] dk \tag{51}$$

where $\omega_\alpha(k)$ is the value of the periodic spectrum of the band $\alpha$, in agreement with Eq. (30) from the main text. Notice that for the frequency denoted by the red star in Fig. 3c, we have $\nu(\omega) = -1$. However, for the same point in Fig. 7c (which lies in the red region to the right of the origin), we have $\tilde{\nu}(\omega) = 1$. To see the relationship between these quantities, notice that $\tilde{\nu}(\omega)$ in Eq. (45) counts the number of zeros of $f(k) = \det[\boldsymbol{D}(k) - \omega]$ in the upper half plane. However, in computing $\nu$ from Eq. (49), the interior of $S$ contains not only the zeros of $F(\lambda) \equiv \det[\boldsymbol{D}(-i\log\lambda) - \omega]$ but also a set of poles. Each of these poles physically represents a boundary condition that arises when transitioning between a Laurent operator to a Toeplitz operator (see Supplementary Note 1 for details). The number of poles depends on the precise discretization of the continuum equations. Hence the winding number $\nu(\omega)$ as given by Eq. (30) represents the difference between the number of zeros (candidate modes) and the number of poles (boundary conditions). Therefore, one should compare $\nu(\omega)$ to $\tilde{\nu}(\omega) - \gamma$ for an appropriate choice of $\gamma$. In practice, $\gamma$ is determined by physically interpreting the boundary conditions implied by the discretization. In Supplementary Note 1, we show that the discretization used in the discrete model and the COMSOL simulations enforce displacement-free boundary conditions and hence corresponds to $\gamma = 2$. Hence $\tilde{\nu}(\omega) - \gamma = \nu(\omega) = -1$ as required for physical consistency.

**Calculation of penetration depth**. In Fig. 3b, we show the penetration depth for modes associated with positive real $\omega$. We can compute this expression by solving $\omega_+(k) = \omega$ in Eq. (28) for complex $k$. In particular, substituting $k = q + i\kappa$ into Eq. (28) yields:

$$0 = 2k\kappa + \frac{P}{2\mu}(k^3 - 3k\kappa^2) \tag{52}$$

$$\omega = \sqrt{\frac{B}{\rho}\left[\left(1 - \frac{P\kappa}{\mu}\right)(k^2 - \kappa^2) - \frac{2P}{\mu}k^2\kappa\right]} \tag{53}$$

Solving Eqs. (52, 53) to leading order in $k$, we obtain Eq. (29).

## Data availability
All data needed to evaluate the conclusions in the paper are present in the paper and/or the Supplementary Information. Extended data and materials in the main text and the Supplementary Information are available upon request by contacting the corresponding authors.

## Code availability
The computer code and algorithm that support the findings of this study are available from the corresponding author upon reasonable request.

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

## Acknowledgements

The authors gratefully thank Prof. Hussein Nassar from University of Missouri for valuable discussions, and Michele Fossati for a critical reading of the manuscript. This work is supported by the Air Force Office of Scientific Research under Grant No. AF9550-18-1-0342 and AF 9550-20-0279 with Program Manager Dr. Byung-Lip (Les) Lee and the Army Research Office under Grant No. W911NF-18-1-0031 with Program Manager Dr. Daniel P Cole. V.V. was supported by the Complex Dynamics and Systems Program of the Army Research Office under grant W911NF-19-1-0268. C.S. was supported by the National Science Foundation Graduate Research Fellowship under Grant No. 1746045. Some of us benefited from participation in the KITP program on Symmetry, Thermodynamics, and Topology in Active Matter supported by Grant No. NSF PHY-1748958.

## Author contributions

Y.C. and G.H. conceived the concept; X.L. conducted experiments; Y.C. and C.S. performed theoretical investigations; Y.C. performed numerical investigations; G.H. and V.V. supervised the research; all the authors discussed the results; all authors wrote the manuscript and interpreted the results.

## Competing interests

The authors declare no competing interests.
