## [Peer Review File · Nature Communications]

REVIEWER COMMENTS

Reviewer #1 (Remarks to the Author):

Authors present a consistent work on what they call "Self-sensing metamaterials with odd micropolarity". The results appear correct and novel.

Several points should be answered before publication:

1. The title is very misleading to my taste and I would find it natural to have the words metamaterial and reciprocity in the title.
2. The odd elasticity is not something very common and I would recommend to the authors to better justify this aspect than adding this sentence-reference 38.
3. In figure 1, I find the deformations in the panel f unclear and would recommend a false color plot from red to blue such that a reader can use it.
4. I do not understand the panel g of the figure 1.
5. Other works have been published using piezo patches to control vibrations and I would recommend to authors to cite them. They are cited in the review paper <https://www.nature.com/articles/s42254-018-0018-y#ref-CR200>

and could be also found on the webpage of Morvan Ouisse:
<https://scholar.google.fr/citations?user=3VPRDFsAAAAJ&hl=fr>

Best regards

Muamer Kadic

Reviewer #2 (Remarks to the Author):

This manuscript presents the design of an active elastic beam with piezoelectric elements (sensors and actuators) that leads to a unidirectional wave amplification/attenuation. The mechanism relies on coupling the bending and shearing motions of the beam to induce an active non-reciprocal modulus.

I find the manuscript well-written and targeted to a broad readership. The topic of this research will be especially relevant in the field of active metamaterials. However, I have the following questions/concerns before I can recommend this manuscript for publication in Nature Communications:

(1) Though the experimental technique is quite interesting and novel, the findings of this work are very close to the ones reported in Ref. 7. The authors distinguish this work by stating that the system is freestanding. What concerns me is that this sounds like an incremental improvement over an existing test setup.

(2) However, I find that the introduction of the beam model with two degrees-of-freedom in non-Hermitian physics is interesting, and it could grant this work enough novelty. But, the presented results are not clear to me. For example, since the beam model has two degrees-of-freedom, I expect two different dispersion branches in the system. In the operational frequency range, does the flexural mode dominate? If yes, it is desirable to support this argument with some analysis. Moreover, since for a frequency range, one may see multiple dispersion branches and mixed modes, will the topological index in Eq. 3 be valid? What happens to the bulk-boundary correspondence? Do authors also observe shear-dominated localized modes at some point?

(3) It is not clear to me why open boundary conditions are needed to witness the localized vibrational modes in the system.

In addition, the following minor comments might help the authors to revise the draft:

(4) In the introduction, the authors state, "Nonetheless, all the active metamaterials so far realized...". It is better to be precise and say "Nonetheless, all the active non-reciprocal metamaterials so far realized..."

(5) x, y, z axes labels are nowhere shown in the images of the main text.

(6) In section IIB, the authors state, "This beam differs substantially from standard energy-absorbing materials in that the amount of energy absorbed is independent of the rate of deformation." Doesn't it contradict the statements later since P is dependent on frequency, and thus, dependent on the rate of deformation?

(7) In section IIC, the authors state, "As a result, the spectrum forms an arc in the complex plane that does not retrace itself (Figure 2a)." In my view, it is better to show an axis of wavenumber to clarify this point at least in the supplementary document.

(8) What is the value of π (material constant) mentioned just before Eq. 5?

Reviewer #3 (Remarks to the Author):

I have carefully reviewed the manuscript "Self-sensing metamaterials with odd micropolarity" by Y. Chen, X. Li, C. Scheibner, V. Vitelli, and G. Huang, which describes the odd elasticity of an active metabeam. The authors use localized active feedback loops to realize a metamaterial that preserves linear and angular momentum but violates Maxwell-Betti reciprocity, which leads to non-Hermitian skin effect.

The results presented in this manuscript are very interesting but need substantial revision to be published in Nature communication. I have several comments that need to be addressed before considering it for publication.

Major comments:

1) Previous works have already observed the emergence of the non-Hermitian Skin effect in active metamaterials (e.g. [7]) and odd elasticity has already been described in previous publications (e.g. [27]). Therefore, the main achievement of this study is the use of micropolar solids to realize a deformable active metamaterial that conserves linear and angular momentum and shows odd elasticity and localization of vibrational modes to sample boundaries. While this is by itself a significant achievement, the authors should better explain what this brings to future studies and applications. For example, they refer to active synthetic biofilaments and membranes in the abstract and in the conclusion, but do not explain what this work is bringing to those fields.

2) In the second to last paragraph of the first page, the authors claim that previous experimental realizations "require the presence of background sources of linear or angular momentum" and as a result "fundamentally require the sample to be in contact with an additional medium that acts as a momentum sink or source". It is not clear to me how these limitations are more significant than the required "independent voltage sources functioning as reservoirs of energy" presented in this paper.

3) The Fig. 2f is a good schematic of the work induced by the coupling between bending and shear. However, the figure describes 4 different steps of deformation where bending only occurs, then a state mixing bending and shear, then shear only and finally back to the initial position. Does it exactly correspond to what happens experimentally? If not, this should be enhanced to better reflect the experiment.

4) In the last paragraph of Section C, the authors discuss the possibility for the beam to absorb mechanical energy and convert it into electric energy. This discussion would bring much more to the paper if the authors would quantify the efficiency of those energy transfer and relate them to eventual applications.

5) In Fig. 4, the authors show that waves propagating in the metabeam are unidirectionally amplified. This observation is not discussed enough. What are the limitations of this unidirectional amplification? What is expected for a larger number of unit cells?

6) The curves in Fig. 4d and the parameters on which they depend should be discussed in more details in the main text.

7) The authors claim that their setup can be easily scaled, but do not provide any argument to corroborate their claim. If the setup is indeed easily scalable, what are the limits of this scalability? Does the electronics showed in Fig.1d, or the use of piezoelectric components affect that scalability?

Minor comments:

1) Active metamaterials have been described with many keywords (active, robotic, smart, autonomous, etc.). I don't think we need more of them. Thus, I suggest the authors to change their title to "active metamaterials with odd micropolarity" or to something similar relating to active metamaterials.

2) The notions of freestanding beam and self-standing beam need to be clarified. In particular, the sentences: "The resulting active metabeam is freestanding-it obeys Newton's third law by preserving both angular and linear momentum." and "Here, we report the design, construction, and experimental demonstration of a self-standing metamaterial".

3) The notion of topological index should be directly referred to in Section C.

4) While this is not a central part of the paper and should not particularly be discussed in the main text, I am curious to know why to authors chose to study 9 unit cells and not more of them.

5) The measurements in Fig 4b is not as smooth as in Fig. 4c. Is this related to the limited accuracy of the measurement method?

Response to the Reviewers' Comments

Reviewer 1

The authors would like to thank Muamer Kadic for his insights and suggestions. We have responded to the comments below, and the paper has been modified accordingly. All the revised and added parts have been highlighted in blue in the revised manuscript and Supplementary Information.

Authors present a consistent work on what they call “Self-sensing metamaterials with odd micropolarity”. The results appear correct and novel.

Response: We thank the reviewer for the positive comments and suggestions on the manuscript. We have responded to the comments in a point-by-point manner below.

Several point should be answered before publication:

1. The title is very misleading to my tast and I would find natural to have the words metamaterial and reciprocity in the title.

Response: Integrating the comments from reviewers 1 and 3, we have modified the title of the manuscript to be “Realization of active metamaterials with odd micropolar elasticity.”

2. The odd elasticity is not something very commun and i would recommend to the authors to better justify this aspect than adding this sentence-reference 38.

Response: Thank you for this suggestion. To improve the manuscript in this regard, we have now added a designated section titled “Energy cycles”, including equations (13) through (18), to directly explain the notion and mathematical formulation of odd elasticity into the main text, rather than relying on external reference [38].

3. In figure 1, I find the deformations in the panel f unclear and would recoment a falscolor plot from red to bleu such that a reader can use it.

Response: We have modified the color scheme in previous Figure 1f (now Figure 2a), for improved clarity.

4. I do not uderstant the panel g of the figure 1.

Response: Thank you for the feedback. To allow more room for explanation, we have now split previous figure 1 into two figures. Previous Fig. 1g is now Fig. 2b. In addition, we have modified the figure, caption, and the corresponding text for improved clarity. Current Figure 2b shows the result of a COMSOL simulation in which a single unit cell is subject to applied deformations at the boundaries. As the unit cell is quasistatically deformed, we measure the work done by the reaction forces on the unit cell. When the deformations are performed in a clockwise direction in strain space, as shown in Figure 2b, the cumulative work done is negative once the unit cell returns to its initial configuration. Hence, energy flows from the external agent into the internal power reserves of the medium. When the cycle is reversed, so is the flow of energy. The ability to inject or extract mechanical energy through quasistatic cycles is synonymous with odd elasticity, i.e. a quasistatic stress-strain relationship that does not follow from a potential energy.

We have added the discussion in Sec. IIC.

5. Others works have been published using piezzo patches to control vibrations and i would recommend to authors to cite them. They are cited in the review paper

<https://www.nature.com/articles/s42254-018-0018-y#ref-CR200>

and could be also found on the webpage of Morvan Ouisse:

<https://scholar.google.fr/citations?user=3VPRDFsAAAAJ&hl=fr>

Response: The papers mentioned above are now properly cited in the revised manuscript, including Refs. [37-46].

Reviewer 2

The authors would like to take this opportunity to thank the reviewer for their insights and suggestions. We have responded to the comments below, and the paper has been modified accordingly. All the revised and added parts have been highlighted in blue in the revised manuscript and Supplementary Information.

This manuscript presents the design of an active elastic beam with piezoelectric elements (sensors and actuators) that leads to a unidirectional wave amplification/attenuation. The mechanism relies on coupling the bending and shearing motions of the beam to induce an active non-reciprocal modulus.

I find the manuscript well-written and targeted to a broad readership. The topic of this research will be especially relevant in the field of active metamaterials. However, I have the following questions/concerns before I can recommend this manuscript for publication in Nature Communications:

Response: We thank the reviewer for the positive comments on the manuscript.

(1) Though the experimental technique is quite interesting and novel, the findings of this work are very close to the ones reported in Ref. 7. The authors distinguish this work by stating that the system is freestanding. What concerns me is that this sounds like an incremental improvement over an existing test setup.

Response: Thank you for the comments. Indeed, there are common physics interest regarding non-reciprocal wave propagation in active metamaterials using local control loops in both Ref. 7 and the current manuscript. However, the working principles and underlying mechanics of the two studies are fundamentally different, which is summarized as:

- (a) In Ref. 7, the nonreciprocal robotic metamaterial is formulated by considering individual bonds that act as “asymmetric springs,” which exert a stronger force on one particle than the other. Within this interaction, linear momentum (or angular momentum, if implemented with torques on rotational degrees of freedom) is not conserved. This implies that to realize such a system, the metamaterial itself must be in contact with a source of linear or angular momentum, such as a substrate or a background fluid. By contrast, the current manuscript suggests a distinct approach in which the material conserves both linear and angular momentum, indicating that it is mechanically freestanding. This liberates the material from the fundamental need to be in contact with a momentum sink. To achieve unidirectional wave amplification in our system, we do not rely on unbalanced forces to break the left-right symmetry, but rather on a micropolar degree of freedom φ which is antisymmetric under parity $\varphi(x) \mapsto -\varphi(-x)$. We build a feedforward control loop inducing asymmetric bend and shear coupling. This approach allows us to achieve non-Hermitian band topology in a system whose effective theory obeys Newton’s third law.
- (b) This conceptual difference at the level of microscopic mechanism is reflected in the emergent continuum mechanics. In Ref. 7, the continuum equations are described by a single component displacement field, whose equation of motion schematically takes the form:

$$\ddot{u} \propto \epsilon \partial_x u + k \partial_x^2 u \quad (1)$$

where u is the displacement and the $\epsilon \partial_x u$ term results in the parity violation. Notice that the right-hand side of Eq. (1) cannot be cast as the divergence of a stress $\partial_x \sigma$ such that σ only depends on deformation, i.e. gradients of u . In our medium, we utilize two coupled fields φ and h intrinsic to the mechanics of a thick beam. In this case, the equations of motion are constructed from conservation laws

$$\rho \ddot{h} = \partial_x \sigma_{zx} \quad (2)$$

$$I \ddot{\varphi} = \partial_x M + \sigma_{zx} \quad (3)$$

Here, σ_{zx} and M are the stress and moment that only depend on the two independent modes of deformation $s(x) = \partial_x h - \varphi$ and $b(x) = \partial_x \varphi$. As noted in section S1A of the S.I, $\varphi = \partial_z u_x$, where $u_x(x, y, z)$ is the x component of the underlying displacement field of the three-dimensional beam. Hence the lowest order derivatives appearing in Eq. (2) and Eq. (3) are second derivatives of the underlying displacement field. Using plane wave assumptions, the two coupled equations can be simplified to a single equation (see Eq. (28) in the main text). The parity violation in our medium comes from a cubic term k^3 in the dispersion, rather than a linear term k that would arise from Eq. 1. (Here, $k = -i\partial_x$ is wavenumber). This is crucial for a medium that conserves linear (or angular) momentum because the momentum must be the divergence of a stress, and the stress itself must be proportional to gradients of the displacement.

We clarified and highlighted the two fundamental differences in Introduction and Sections IIA and IID.

- (2) However, I find that the introduction of the beam model with two degrees-of-freedom in non-Hermitian physics is interesting, and it could grant this work enough novelty. But, the presented results are not clear to me. For example, since the beam model has two degrees-of-freedom, I expect two different dispersion branches in the system. In the operational frequency range, does the flexural mode dominate? If yes, it is desirable to support this argument with some analysis. Moreover, since for a frequency range, one may see multiple dispersion branches and mixed modes, will the topological index in Eq. 3 be valid? What happens to the bulk-boundary correspondence? Do authors also observe shear-dominated localized modes at some point?

Response: Thanks for recognizing another scientific merit of the manuscript. Indeed, the continuum model has two independent degrees of freedom: the transverse displacement $h(x)$ and the rotation angle $\varphi(x)$. Since the equations are inertial, there is a total of four bands. We have added Figure M1, which now clearly show all four bands in the continuum model. The ungapped bands are the flexural modes, which are dominated by transverse displacement. In principle, the continuum theory predicts two gapped bands that are dominated by shearing. However, we emphasize a crucial difference between the gapped and ungapped bands. For the ungapped (flexural) bands, by definition $\omega(k) \rightarrow 0$ as $k \rightarrow 0$. Hence, one expects that there is a nonzero, but empirically determined, range of validity for the continuum theory at small k . For the gapped band, ω attains a finite value at $k = 0$, and hence one must explicitly check (either with numerics or with experiments) whether or not the continuum theory is successful at that frequency scale. We note that our experiments probe frequencies $\leq 10\text{kHz}$, while the band gap is at $\omega_1 \approx 100\text{kHz}$, and hence we expect that only the flexural modes are relevant in experiments.

In our case, we both expect and numerically confirm that the predictions of the continuum theory are not applicable in the frequency range of ω_1 . Our continuum theory is formulated in terms of h and φ , which are the key degrees of freedom to leverage the active feedback by stretching and compressing the piezoelectrics. However, there are certainly additional modes of deformation: a torsional, longitudinal, and out of plane motion of the beam. Our approach is predicated on the assumption that one can safely simplify the analysis by ignoring these additional modes. To further validate this assumption, we have added Fig. S2 in the S.I., in which we perform finite element COMSOL simulations for the 15 lowest modes of the beam. As anticipated, for frequencies less than 7kHz only the flexural band has a significant imaginary part (Figure S2). Moreover, for frequencies less than 2.5kHz we see quantitative agreement between the theory and numeric with no fitting parameters (Figure 3a).

Furthermore, in the Methods and S.I. section S1B, we have provided a more detailed explanation of the connection between the winding number $\nu(\omega)$ and the boundary modes. As described in the revised methods, the continuum winding number is given by

$$\nu(\omega) = \lim_{R \rightarrow \infty} \frac{1}{2\pi i} \oint_{\Gamma(R)} \frac{d}{dk} \log \det[D(k) - \omega] dk \quad (4)$$

where $\Gamma(R)$ is a counterclockwise semicircle of radius R in the upper half of the complex k plane. Qualitatively, Eq. (4) counts the number of times that the complex spectrum winds around a frequency ω . The value of this winding number depends on which degrees of freedom, such as the shear mode, are present in the continuum analysis. We now more clearly explain in Section S1B of the S.I.

“...the physically relevant piece of information is not necessarily the absolute value of $\nu(\omega)$, but rather the relative value $\nu(\omega') - \nu(\omega'')$ for any two given frequencies ω' and ω'' . Suppose, for example, there are n left localized modes at ω' with a given set of boundary conditions. The absolute value of $\nu(\omega')$ and the way in which the boundary conditions are quantified will depend on the details of the continuum theory. However, given only the value of $\nu(\omega') - \nu(\omega'')$, one can conclude that there will be $n - \nu(\omega') + \nu(\omega'')$ left localized modes at ω'' for the same set of boundary conditions. Finally, we note that to compute the relative value $\nu(\omega') - \nu(\omega'')$, one need only draw a line in the complex plane from ω'' to ω' and count the number of signed crossings with the periodic boundary spectrum. Hence, the value $\nu(\omega') - \nu(\omega'')$ only depends on accurately resolving the spectrum in the frequency range of interest, and not on features of the spectrum outside the range of validity of the theory.”

See the methods and section S1B of the S.I. for further discussion.

- (3) It is not clear to me why open boundary conditions are needed to witness the localized vibrational modes in the system.

Response: We have clarified this point in the manuscript. Indeed, one could witness the unidirectional wave amplification in a system with periodic boundary conditions. As we now explain in the manuscript, the periodic boundary spectrum on its own implies unidirectional wave amplification:

“The simultaneous breaking of parity and energy conservation allows our active micropolar metamaterial to selectively amplify and attenuate waves based on their direction of travel. ... As illustrated in Figure 3a, for $P > 0$, $\text{Im}(\omega) > 0$ whenever $\text{Re}(d\omega/dk) > 0$ and $\text{Im}(\omega) < 0$ whenever $\text{Re}(d\omega/dk) < 0$. Physically, this means that wave packets traveling to the right are amplified, while wave packets traveling to the left are attenuated.”

Moreover, we now explain in the Methods that the difference between systems with open and periodic boundary conditions is whether the differential operator that defines the equations of motion allows formal eigenvectors with complex k . With periodic boundary conditions the wave number k is constrained to be real. However, in a semi-infinite system $x \in [0, \infty)$, one can allow solutions that decay to the right $e^{(-\text{Im } k + i \text{Re } k)x}$ while maintaining a finite L^2 norm (representing the mechanical energy). The winding number $\nu(\omega)$ counts the number of such decaying modes at a given complex frequency. However, the bulk physics (far away from the boundary) is unchanged by a change in boundary condition since the qualitative physics is already evident from the periodic boundary spectrum.

In addition, the following minor comments might help the authors to revise the draft:

- (4) In the introduction, the authors state, “Nonetheless, all the active metamaterials so-far realized...”. It is better to be precise and say “Nonetheless, all the active non-reciprocal metamaterials so-far realized...”

Response: We took this suggestion and changed the manuscript accordingly.

- (5) x , y , z axes labels are nowhere shown in the images of the main text.

Response: We added x , y , z axes labels to Fig. 1e.

- (6) In section IIB, the authors state, “This beam differs substantially from standard energy-absorbing materials in that the amount of energy absorbed is independent of the rate of deformation.” Doesn’t it contradict the statements later since P is dependent on frequency, and thus, dependent on the rate of deformation?

Response: We have revised the discussion as

“Notice that the work done per cycle can be either positive or negative depending on the orientation of the path, and is independent of the rate of the process (in the quasistatic limit).”

- (7) In section IIC, the authors state, “As a result, the spectrum forms an arc in the complex plane that does not retrace itself (Figure 2a).” In my view, it is better to show an axis of wavenumber to clarify this point at least in the supplementary document.

Response: To clarify this, we have added Figs. M1 and S2 to the Methods and S.I. Furthermore, current Fig. 3a (previously Fig. 2a), now shows the periodic boundary spectrum with hue indicating wave number k to help provide this information.

- (8) What is the value of Π (material constant) mentioned just before Eq. 5?

Response: For this design, $\Pi = 4.7 \times 10^6 \text{N/m}^3$.

Reviewer 3

The authors would like to take this opportunity to thank the reviewer for their careful review. We have responded to the comments below, and the paper has been modified accordingly. All the revised and added parts have been highlighted in blue in the revised manuscript and Supplementary Information.

I have carefully reviewed the manuscript “Self-sensing metamaterials with odd micropolarity” by Y. Chen, X. Li, C. Scheibner, V. Vitelli, and G. Huang, which describes the odd elasticity of an active metabeam. The authors use localized active feedback loops to realize a metamaterial that preserves linear and angular momentum but violates Maxwell-Betti reciprocity, which leads to non-Hermitian skin effect. The results presented in this manuscript are very interesting but need substantial revision to be published in Nature communication. I have several comments that need to be addressed before considering it for publication.

Response: We thank the reviewer for the positive comments on the manuscript.

Major Comments:

- 1) Previous works have already observed the emergence of the non-Hermitian Skin effect in active metamaterials (e.g. [7]) and odd elasticity has already been described in previous publications (e.g. [27]). Therefore, the main achievement of this study is the use of micropolar solids to realize a deformable active metamaterial that conserves linear and angular momentum and shows odd elasticity and localization of vibrational modes to sample boundaries. While this is by itself a significant achievement, the authors should better explain what this brings to future studies and applications. For example, they refer to active synthetic biofilaments and membranes in the abstract and in the conclusion, but do not explain what this work is bringing to those fields.

Response: We thank the reviewer for this suggestion. The abstract now states:

“Our continuum approach, built on symmetries and conservation laws, could be exploited to design others systems such as synthetic biofilaments and membranes with feed-forward control loops.”

In the revised manuscript, we now explain more clearly the features of our approach that lend generalizability to future studies and applications. First, our mechanical approach is predicated on the notion of a feedforward control loop, a generic concept that can exist in both metamaterial and biological contexts. By invoking a feedforward control loop, one can generically break the symmetry between deformation and response known as Maxwell-Betti reciprocity. Second, we note that our theoretical approach is grounded in the framework of hydrodynamics and hence relies not on the details of the microscopic construction, but rather on the broken symmetries and conservation laws. The engineered modulus we present (which is representative of generalized moduli that could also arise in membranes) requires only the breaking of appropriate symmetries (in this case parity and Maxwell-Betti reciprocity). This grounding in continuum theory is what makes our approach especially generalizable to the mechanics in other systems.

We have also highlighted the generalizability in Discussion.

- 2) In the second to last paragraph of the first page, the authors claim that previous experimental realizations “require the presence of background sources of linear or angular momentum” and as a result “fundamentally require the sample to be in contact with an additional medium that acts as a momentum sink or source”. It is not clear to me how these limitations are more significant than the required “independent voltage sources functioning as reservoirs of energy” presented in this paper.

Response: Thank you for the comments. We first emphasize that our goal is only to point out the different properties and requirements for the two approaches, with no intention to address which one is better. The strategy in Ref. 7 is to engineer interactions that can exert unbalanced forces between two sites. Thus, this approach inherently breaks the linear momentum conservation (Newton’s third law). Therefore, such a system needs to be grounded or be in contact with an additional medium that acts as a momentum sink or source (e.g. a substrate or background fluid). By contrast, the current active system is realized by apply opposite shear stresses proportional to the bending curvature to induce asymmetric bending-shear coupling. Therefore, the current system only needs to be connected to an independent voltage, or more generally energy, source. Beyond the conceptual distinction, one practical difference is that a voltage or power source, unlike a source of linear momentum, can more easily be housed inside the medium itself. For example, the energy source can be provided in the form of batteries that do not require internal moving parts. (See also our answer to the first question by the second reviewer).

We have added this discussion in Sec. IIA.

- 3) The Fig. 2f is a good schematic of the work induced by the coupling between bending and shear. However, the figure describes 4 different steps of deformation where bending only occurs, then a state mixing bending and shear, then shear only and finally back to the initial position. Does it exactly correspond to what happens experimentally? If not, this should be enhanced to better reflect the experiment.

Response: Thank you for the comment. In the revised manuscript, we now clarify that the formula

$$\text{Work} = P \times \text{Area} \quad (5)$$

applies in the quasi-static limit, and therefore it alone does not directly describe the experiments. Rather, the conceptual purpose of this figure is to illustrate that the modulus P does not derive from a potential energy, even in the quasi-static limit in which rate-dependent effects (e.g. dissipation) are irrelevant. We note that in the quasi-static limit, Eq. (5) is valid regardless of the precise geometry of the cycle, and hence the square is representative of a more general scenario.

To enhance the discussion, we now include the following text in the Methods:

“To gain intuition on the elastodynamics of the odd micropolar metabeam, it is useful to consider the notion of a cycle at finite frequency. At a finite frequency ω , the modulus P need not be real and we may write $P = |P|e^{i\phi_P}$. In this case, both the real and imaginary parts of P will contribute to the energy extracted. For example, consider a cyclic protocol which involves bending of amplitude $|b|$ and shearing of amplitude $|s|$ and relative phase delay ϕ_B . Applying Eq. (M3), the total energy extracted per cycle is

$$\text{Work} = -\pi|P||s||b|\sin(\phi_B + \phi_P) \quad (4)$$

Notice that Eq. (4) gives mechanistic insight into the amplification of the waves observed in the experiment. When P is nonzero, the eignemodes of $D(k)$ comprising a given plane wave will generically have a phase delay between bending and shearing. Hence, after one cycle, the nonconservative stresses will have converted stored electrical energy into mechanical energy. This conversion will cause the amplitude of the eigenvector to grow in proportional to its current amplitude, for which $|s||b|$ is a proxy. Hence, the mode will be exponentially amplified, as reflected by the imaginary component of the eigenfrequencies.”

- 4) In the last paragraph of Section C, the authors discuss the possibility for the beam to absorb mechanical energy and convert it into electric energy. This discussion would bring much more to the paper if the authors would quantify the efficiency of those energy transfer and relate them to eventual applications.

Response: For quasi-static deformations, the amount of electrical energy transferred to the mechanical domain, or vice versa, is controlled by the equation $\text{Work} = P \times \text{Area}$, which generalizes to $\text{Work} = -\pi|P||s||b|\sin(\phi_B + \phi_P)$ at finite frequency. The power is then the work per cycle divided by the period of the cycle Work/T . The optimization problem then resembles that of a wind turbine, for which the resistance of the wind turbine must be tuned to maximize the product of the number of cycles per second and the power extracted per cycle.

While the optimal design will likely be task and application dependent, we now provide an illustration of the efficiency in our current system at absorbing the energy flux of incident waves. In the S.I., we now provide a characterization of energy absorption efficiency:

“In addition to quasistatic strain controlled deformations, we also examine the efficiency of our metabeam at absorbing energy from finite frequency waves. To do so, we first perform numerical simulations matching the parameters in our experiments. From these numerical tests, we measure the mechanical energy flux from the left and right boundaries of a given unit cell (in this case the fifth unit cell). Denoting these energy fluxes by F_L and F_R , respectively, we define the energy absorption efficiency of left-traveling waves by $\mathcal{E} = (F_R - F_L)/F_R$, where $|F_R| > |F_L|$. At 2 kHz, we find the peak absorption efficiency per unit cell is equal to $\mathcal{E} = 0.38$. Figure S3 shows this efficiency at frequencies from 0.6 to 3.0 kHz, where the waves propagating from the right to the left are attenuated.”

- 5) In Fig.4, the authors show that waves propagating in the metabeam are unidirectionally amplified. This observation is not discussed enough. What are the limitations of this unidirectional amplification? What is expected for a larger number of unit cells?

Response: While the linear properties of the unidirectional waves is captured by linear theory discussed in the manuscript, there indeed exist additional features of the experiment that in principle go beyond this effect. In experiments, the unidirectional amplification can be limited by the maximum output voltage ($\pm 45\text{V}$) of our electric control system. In practice, to maximize the amplification ratio, one strategy is to use a modest value of the transfer

function, say $|H_0| \approx 3$, and increase the number of unit cells for this purpose over which the wave is amplified. In addition, we also need to consider stability conditions of the metamaterial in experiments. In particular, we find that small feedback effects emerge within individual unit cells associated with imperfections in fabrication. When $|H_0|$ exceeds a critical value, the antisymmetric actuating voltages can no longer produce zero sensing signals in experiments and the system experiences an instability. This critical value is $|H_0| \approx 6$ in our current system, but it is highly dependent on the fabrication details. Due to the self amplification of the beam, the critical value of $|H_0|$ usually decreases as the number of unit cells increases. Aside from effects related to electronic feedback, we note that a wave propagating along a beam with many unit cells cannot be described indefinitely by the linear theory. At some point, when the strains are no longer small, a fully nonlinear theory would be necessary to model the phenomena.

We have added the discussion in Sec. IIF and S.I..

- 6) The curves in Fig. 4d and the parameters on which they depend should be discussed in more details in the main text.

Response:

We now discuss the curves in Fig. 4d (now Fig. 5d) in more details in the main text.

“... As described in the S.I., the solid theoretical curves are produced using a semi-analytical technique known as the transfer-matrix method. The transfer-matrix method utilizes the beam geometry, known material parameters, and electronic feedback measured in simulations. No fitting parameters are used in the comparison between experiment and the transfer-matrix method curve.

As illustrated in Figure 5f, the transfer function $H(\omega)$ is chosen such that $-\pi/2 < \arg(P) < 0$ when $\omega < \omega_0$. In this case, we find $\kappa < 0$ for both left- and right-propagating waves (panel e). A value of $\kappa < 0$ implies that waves propagating to the left are attenuated whereas waves propagating to the right are amplified, which is confirmed by the FFT intensity in panel d. Likewise $-\pi < \arg(P) < -\pi/2$ for $\omega > \omega_0$. In this case, we find $\kappa > 0$, indicating that waves propagating to the right are attenuated whereas waves propagating to the left are amplified. In addition, when $\omega = \omega_0$, $\arg(P) = -\pi/2$, and the waves propagating to the left and right display no attenuation or amplification. For this case, the metamaterial resides on a pseudo-Hermitian point, and the differences between left- and right-propagating waves reside only in the wavelengths and phase velocities.”

- 7) The authors claim that their setup can be easily scaled, but do not provide any argument to corroborate their claim. If the setup is indeed easily scalable, what are the limits of this scalability? Does the electronics showed in Fig.1d, or the use of piezoelectric components affect that scalability?

Response: We claim that our setup can be easily scaled because the current active system is constructed on a linear continuous medium and electronically controlled elements can be easily miniaturized with the advance of MEMS fabrication technique. We now cite additional literature relevant for the miniaturization of the electronic and mechanical components, see e.g. Refs. [14,65,66]. We anticipate that when pushing the boundaries for small scales, a limiting factor will be the fabrication methods needed to coat the piezoelectric components to host structures. When the system is applied in large structure scales, one will need to engineer control elements with enough power to generate the required stresses. To facilitate the estimation of wave phenomena at different scales, we now also cast the wave equation [Eq. (23)] in terms of characteristic lengths $\ell_1 \equiv \sqrt{I/\rho}$, $\ell_2 \equiv \sqrt{B/\mu}$, a characteristic frequency $\omega_1 \equiv \sqrt{\mu/I}$, and the normalized magnitude of the odd modulus $\tilde{P} \equiv P/\sqrt{B\mu}$, which can be compared across platforms.

Minor Comments

- 1) Active metamaterials have been described with many keywords (active, robotic, smart, autonomous, etc.). I don't think we need more of them. Thus, I suggest the authors to change their title to “active metamaterials with odd micropolarity” or to something similar relating to active metamaterials.

Response: Integrating the comments from reviewers 1 and 3, we have modified the title of the manuscript to be “Realization of active metamaterials with odd micropolar elasticity.”

- 2) The notions of freestanding beam and self-standing beam need to be clarified. In particular, the sentences: “The resulting active metabeam is freestanding-it obeys Newton's third law by preserving both angular and linear momentum.” and “Here, we report the design, construction, and experimental demonstration of a self-standing metamaterial”.

Response: We have made the usage consistent throughout the manuscript by adopting the phrase “freestanding”.

3) The notion of topological index should be directly referred to in Section C.

Response: We have revised this section for enhanced clarity.

4) While this is not a central part of the paper and should not particularly be discussed in the main text, I am curious to know why to authors chose to study 9 unit cells and not more of them.

Response: The number of unit cells would not change any conclusions of the manuscript. In experiments, we found 9 unit cells are sufficient to demonstrate expected phenomenon. As mentioned in previous response, increasing the number of unit cells will make the control system more complex.

5) The measurements in Fig 4b is not as smooth as in Fig. 4c. Is this related to the limited accuracy of the measurement method?

Response: Yes. In Fig. 4b, the the signal to noise is decreased due to the attenuation of the wave.

REVIEWER COMMENTS

Reviewer #1 (Remarks to the Author):

Authors have implemented my suggestions and the paper can be accepted in the current form.

Reviewer #2 (Remarks to the Author):

I find the revised manuscript addresses most of my concerns. I appreciate the efforts put by the authors in clarifying many subtle issues. However, the following points in the revised version are still not clear to me, and I encourage the authors to consider them:

1. In the revised introduction, the way the authors have presented the two notions of reciprocity is puzzling for me. They claim that the second notion (Maxwell-Betti reciprocity) is independent of the first notion (using external forces or torques). In my understanding, the first way too can break the Maxwell-Betti reciprocity, for example, in ref[7]. Therefore, the introduction of an independent second notion is not clear to me. I would interpret the authors' system as the one where "system" is defined differently. This system includes the actuators as opposed to the system in ref[7]. Moreover, I also notice that the systems in refs[7, 29] can be treated as discrete systems, whereas the current system is a continuum. Therefore, I wonder if the authors try to convert their continuous system to a discrete model (springs and masses), they would end up with a system with sources/sinks of linear/angular momentum similar to refs[7, 29].

2. Though I am quite happy with the newly added information on the topological index in Methods, it is not clear to me how the authors deduce eqs.(M6) and (M7) from eq.(M5)? Moreover, the assumption seems to be that one needs "free" boundary conditions. What happens when one has "fixed" boundary conditions. A detailed discussion may not be required at this point, but I encourage authors to write in a few lines why a free boundary is chosen for this framework.

3. I did not understand the discussion after eq.(M11). Why boundary conditions are implicit in the discrete system? I see that the statement is only relevant for a semi-infinite system, which is the case for both the continuum [eq.(M5)] and discrete [eq.(M11)] cases. Therefore, I do not understand why the calculation of edge states should differ in the continuum and discrete systems.

4. I notice that the experimental setup in Fig. 5a would have neither free boundaries nor periodic boundaries. I wonder if this is important, especially in calculating the decay lengths in Fig. 5e for the finite sample with 9 unit cells.

5. Figs. M1c,d are hard to understand. Is there any way to reconcile these with the results shown in Fig. 3? Can ν be -1 in the regions shown in Figs. M1c,d just as the cases shown in Fig. 3? Also, in Fig. M1e, the authors plot localized modes in low frequencies. It will also be interesting to see how these look like in high frequencies, e.g., those lying in region 2 of Figs. M1c,d for both bending and shear dominant branches.

6. In SI, The authors write "Hence, the shear dominated modes should not necessarily be thought of as a physical prediction made by the continuum theory." In Fig. S2a, I wonder if the concave upward branch at $\text{freq} \sim 95$ kHz at $k=0$ is the shear-dominated branch predicted by the continuum theory. For a *small* wavenumber, I do not see why a continuum theory that already accommodates multiple degrees of freedom (displacement and rotation) not predicting this higher branch.

Some minor comments:

7. In Figs.3a,b, x-axes units should be kHz.

8. The authors can write in the caption which method is used to calculate Figs.3c,d.

Reviewer #3 (Remarks to the Author):

I have carefully reviewed the second version of the manuscript "Realization of active metamaterials with odd micropolar elasticity" by Y. Chen, X. Li, C. Scheibner, V. Vitelli, and G. Huang, which describes the odd elasticity of an active metabeam.

The authors have greatly improved their manuscript by explaining in more details some of the new concepts discussed in the paper and by better describing their experimental setup. By doing so, the authors have answered all my previous questions. In particular, the authors now perfectly describe the difference between their current setup and previous publications (ref. [7] of the main text) and demonstrate that these first publications open the way to many more interesting results on the influence of feed-forward control in metamaterials and more. I believe that the manuscript in its current form should be published in Nature Communication and that it will have a significant impact on the different fields studying active systems, in particular active materials and metamaterials.

My only suggestion concerns the reference to application in the control of filaments and membranes arising in biological media in the conclusion of the manuscript. I think that it is important to emphasize more on the significant work that is left to better understand the role of nonlinearities, dissipation processes and disorder to relate both fields.

Response to the Reviewers' Comments

Reviewer 1

Authors have implemented my suggestions and the paper can be accepted in the current form.

Response: Thank you for the encouraging remarks.

Reviewer 2

I find the revised manuscript addresses most of my concerns. I appreciate the efforts put by the authors in clarifying many subtle issues. However, the following points in the revised version are still not clear to me, and I encourage the authors to consider them:

1. In the revised introduction, the way the authors have presented the two notions of reciprocity is puzzling for me. They claim that the second notion (Maxwell-Betti reciprocity) is independent of the first notion (using external forces or torques). In my understanding, the first way too can break the Maxwell-Betti reciprocity, for example, in ref[7]. Therefore, the introduction of an independent second notion is not clear to me. I would interpret the authors' system as the one where "system" is defined differently. This system includes the actuators as opposed to the system in ref[7]. Moreover, I also notice that the systems in refs[7, 29] can be treated as discrete systems, whereas the current system is a continuum. Therefore, I wonder if the authors try to convert their continuous system to a discrete model (springs and masses), they would end up with a system with sources/sinks of linear/angular momentum similar to refs[7, 29].

Response: Thank you for this clarifying question. We agree that it is correct to say that the systems considered in [7,9] violate both Maxwell-Betti reciprocity and reciprocity in the sense of conservation of angular momentum. We have adjusted the text and the placement of the references to make this more clear. We nonetheless claim that the two notions are conceptually distinct. Maxwell-Betti reciprocity is the statement that the linear response of a system is consistent with a potential energy function. To violate Maxwell-Betti reciprocity, a system needs to be in contact with a source of energy. By contrast, for a system to violate linear or angular momentum conservation, it needs to be in contact with an external source of linear or angular momentum. These notions are independent in the sense that a system can violate neither, either one, or both simultaneously. In the S.I. we now provide a minimal toy systems in each category to help clarify this distinction (Section S1B).

The discrete vs continuous analysis does not affect our claim that the metabeam constructed here conserves linear and angular momentum. This claim is rooted in the fact that there is nothing physically attached or coupled to the beam that imparts linear or angular momentum (other than the wires which are mechanically negligible). For our device, the active stresses come from the piezoelectric patches. The reason that we consider the piezoelectric patches as part of our system (rather than as external agents) is that they add mass to the system and they dynamically comove and deform with the beam. In this sense, the active components are part of the modeled beam, as opposed to external sinks of angular or linear momentum.

We have taken the reviewer's suggestion and constructed a simple discrete model of our beam that manifestly illustrates the conservation of angular and linear momentum (See Section IIF of the main text). This toy model is constructed from linear and torsional springs that manifestly exert equal and opposite forces and torques on any two elements which they connect. Indeed, while the total angular and linear momentum is conserved, it is interesting to note that the beam can exhibit a mechanical transfer between "orbital" angular momentum in the transverse motion of the beam and internal "spin" angular momentum stored in the rotation of the beam's cross-section.

2. Though I am quite happy with the newly added information on the topological index in Methods, it is not clear to me how the authors deduce eqs.(M6) and (M7) from eq.(M5)? Moreover, the assumption seems to be that one needs "free" boundary conditions. What happens when one has "fixed" boundary conditions. A detailed discussion may not be required at this point, but I encourage authors to write in a few lines why a free boundary is chosen for this framework.

Response: In the S.I., we now detail the mathematical connection between Eqs. (M6) and (M7). Strictly speaking, free boundary conditions are not necessarily assumed. As we now make more clear, the quantity γ

represents the number of independent, homogeneous boundary conditions on the differential equation. We have add a section to the S.I. (Section S1D), in which we define these terms precisely and we provide examples of how the formalism incorporates stress-free boundary conditions or motion-free boundary conditions.

3. I did not understand the discussion after eq.(M11). Why boundary conditions are implicit in the discrete system? I see that the statement is only relevant for a semi-infinite system, which is the case for both the continuum [eq.(M5)] and discrete [eq.(M11)] cases. Therefore, I do not understand why the calculation of edge states should differ in the continuum and discrete systems.

Response: We have revised the text in the methods section to clarify this point, and we have provided a more detailed derivation in the S.I (Section S1E). In the discrete case, finding the eigenmodes mathematically corresponds to finding the eigenvectors of a Toeplitz operator, which is a semi-infinite matrix. This Toeplitz operator is the truncation of a Laurent operator (representing the infinite system). Qualitatively speaking, the eigenvectors of the Toeplitz operator are formed by linear combinations of eigenvectors of the Laurent operator that must satisfy boundary conditions arising from the truncation. The precise boundary conditions depend on how the continuum equations are discretized. In the S.I. we provide a specific, concrete examples relevant for the system at hand.

We also help clarify the distinction between discrete and continuum by introducing separate notation for the winding number in the continuum $\tilde{\nu}(\omega)$ and in the discrete case $\nu(\omega)$, which are mathematically distinct quantities. The winding number in the continuum is given by:

$$\tilde{\nu}(\omega) = \lim_{R \rightarrow \infty} \frac{1}{2\pi i} \oint_{\Gamma(R)} \frac{d}{dk} \log f(k) dk \quad (1)$$

where $f(k) = \det[\mathcal{D}(k) - \omega]$ is a polynomial with all positive powers of k . By contrast, the winding number in the discrete case is given by:

$$\nu(\omega) = \frac{1}{2\pi i} \oint_{S^1} \frac{d}{d\lambda} \log F(\lambda) d\lambda \quad (2)$$

where $F(\lambda) = \det[\mathcal{D}(-i \log \lambda) - \omega]$ is a polynomial in λ . Here S^1 is the unit circle, $\mathcal{D}(k)$ is the discrete dynamical matrix, and $\lambda = e^{iLk}$ where L is the lattice spacing and k is the wavenumber. Notice that the poles of $f(k)$ are all located at $|k| \rightarrow \infty$ and are therefore not included within the contour $\Gamma(R)$, which consists of $[-R, R]$ closed by $Re^{i\phi}$ with $\phi \in [0, 2\pi]$. Hence the integral in Eq. (1) counts only the zeros of $f(k)$ which represents candidate eigenvectors of the differential equations. For the candidate eigenvectors to become genuine eigenmodes, the eigenvectors must satisfy a set of γ imposed by boundary conditions. In the discrete case, however, $F(\lambda)$ contains both positive and negative powers of λ . The negative powers of λ give rise to poles at $\lambda = 0$, which are included within the contour S^1 . Physically, each of the poles represents a boundary condition imposed by the truncation of the Laurent operator. By the Cauchy argument principle, the integral in Eq. (2) counts the difference between the number of zeros (candidate modes) and the number of poles (boundary conditions), and hence effectively absorbs the γ into the definition of ν itself. Qualitatively, the difference between the continuum and discrete approaches lies in the fact that contour S^1 for the discrete integral includes $\text{Im}[k] = \infty$ (namely at $\lambda = 0$), which encodes the information about boundaries, while the continuum contour $\Gamma(R)$ excludes $\text{Im}[k] \rightarrow \infty$ and so the boundary information must be provided separately.

4. I notice that the experimental setup in Fig. 5a would have neither free boundaries nor periodic boundaries. I wonder if this is important, especially in calculating the decay lengths in Fig. 5e for the finite sample with 9 unit cells.

Response: The theoretical curves in Figs. 5d-e are produced using the transfer matrix method, which would formally apply in the limit of an infinitely long material. In experiments, we fabricated the metamaterial in the middle portion of a host beam. When waves cross the boundaries from host beam to the metamaterial, the reflection at boundaries between the host beam and the metamaterial is negligible, as evidenced by our numerical and experimental results. To suppress reflected waves at the free boundaries of the host beam, we bonded two layers of clay on the host beam with sufficient lengths. This way, waves can propagate through the metamaterial with approximated infinite boundary conditions. The decay length is then obtained by calculating the wave amplitudes at different points in the metamaterial.

5. Figs. M1c,d are hard to understand. Is there any way to reconcile these with the results shown in Fig. 3? Can ν be -1 in the regions shown in Figs. M1c,d just as the cases shown in Fig. 3? Also, in Fig. M1e, the authors

plot localized modes in low frequencies. It will also be interesting to see how these look like in high frequencies, e.g., those lying in region 2 of Figs. M1c,d for both bending and shear dominant branches.

Response: Thank you for this clarifying comment. Let us first begin by clarifying how to read Fig. M1c,d. Equation M5 of the methods provides an expression that defines the winding number $\tilde{\nu}(\omega)$ associated with a frequency ω in the complex plane. Fig. M1c,d provides an illustration of that formula by coloring each region by the value of the winding number: $\tilde{\nu} = 1$ is red, $\tilde{\nu} = 2$ is grey, and $\tilde{\nu} = 3$ is blue. To see visually how this winding number is computed, the number corresponding to a given point ω in the complex plane is the number of times that the solid lines wrap around the point ω . A useful trick for determining the winding number is to draw a line starting from ∞ and ending at the point of interest. Tabulate the number of signed crossings between the drawn line and the spectrum, and this will yield the winding number.

Next, let us address the question regarding modes plotted in panel e. The modes shown in e are purely schematic and illustrate a mode generically decaying to the left or to the right. We have adjusted the caption to emphasize this and we have replaced the label $h(x)$ with $A(x)$ to stand for amplitude. As suggested, in the S.I. (Section S1F) we now explicitly compute the eigenmodes for a frequency ω near the origin and one in the grey region. In addition, we also compute the eigenmodes at $P = 0$ to leading order in k to illustrate why the bands are referred to as bending and shearing dominated in Fig. M1a.

Finally, we note that it is indeed possible to reconcile the winding numbers in Fig. M1c,d with those shown in Fig. 3. Our adjusted notation now clarifies that Fig. 3 involves the discrete winding number $\nu(\omega)$ while Fig. M1c,d involves the continuum winding number $\tilde{\nu}(\omega)$. Following question 3, to convert between the two we need to physically interpret the boundary conditions implied by the discrete model. In this case, the boundary conditions are displacement free, and so we take $\gamma = 2$. Examining the red region near the origin of Fig. M1c, we have $\nu(\omega) = \tilde{\nu}(\omega) - \gamma = 1 - 2 = -1$, which agrees with the winding number around the star in Fig. 3c.

6. In SI, The authors write “Hence, the shear dominated modes should not necessarily be thought of as a physical prediction made by the continuum theory.” In Fig. S2a, I wonder if the concave upward branch at freq 95 kHz at $k=0$ is the shear-dominated branch predicted by the continuum theory. For a *small* wavenumber, I do not see why a continuum theory that already accommodates multiple degrees of freedom (displacement and rotation) not predicting this higher branch.

Response: Thank you for this question. In Fig. S10 of the S.I., reproduced here, we now provide a rendering of each mode at $kL = 0.314$. Upon inspection, we find that modes 10-13 (near 95 kHz) do not resemble the shear dominated mode predicted by the continuum theory. At high frequencies, the beam can access modes with multiple undulations per unit cell. These modes hybridize with the gaped modes predicted by the continuum theory and invalidate the continuum predictions at high frequency. This is why we invoke both the conditions of long wavelength and low frequency as conditions for applying the continuum theory.

Some minor comments:

7. In Figs.3a,b, x-axes units should be kHz.

Response: Thank you, this has been fixed.

8. The authors can write in the caption which method is used to calculate Figs.3c,d.

Response: We have clarified the caption for Fig. 3. Panels c and d are schematics for illustration purposes. The spectral winding in panel c reflect the COMSOL data shown explicitly in panel a for $P > 0$. The growing and decaying wave-forms in d schematically illustrate the localization illustrated in panel b.

Reviewer 3

I have carefully reviewed the second version of the manuscript “Realization of active metamaterials with odd micropolar elasticity” by Y. Chen, X. Li, C. Scheibner, V. Vitelli, and G. Huang, which describes the odd elasticity of an active metabeam.

The authors have greatly improved their manuscript by explaining in more details some of the new concepts discussed in the paper and by better describing their experimental setup. By doing so, the authors have answered all my previous questions. In particular, the authors now perfectly describe the difference between their current setup and previous publications (ref. [7] of the main text) and demonstrate that these first publications open the way to many more interesting results on the influence of feed-forward control in metamaterials and more. I believe that

the manuscript in its current form should be published in Nature Communication and that it will have a significant impact on the different fields studying active systems, in particular active materials and metamaterials.

My only suggestion concerns the reference to application in the control of filaments and membranes arising in biological media in the conclusion of the manuscript. I think that it is important to emphasize more on the significant work that is left to better understand the role of nonlinearities, dissipation processes and disorder to relate both fields.

Response: Thank you for the encouraging remarks. We have modified the discussion to reflect the important future work required to achieve these goal.

FIG. 1: The lowest 16 modes corresponding to bands shown in Fig. 11. The wavenumber is taken to be $kL = 0.314$, where L is the lattice spacing. The odd micropolar modulus is $P = 3\text{II}$. The color bar denotes the z (vertical) component of the displacement field.

REVIEWERS' COMMENTS

Reviewer #2 (Remarks to the Author):

I appreciate the authors in providing very detailed responses to my concerns. I am satisfied with these responses. The revised manuscript can be published as-is.

Reviewer #2 (Remarks to the Author):

I appreciate the authors in providing very detailed responses to my concerns. I am satisfied with these responses. The revised manuscript can be published as-is.

Response: We thank the reviewer for his/her valuable suggestions.